# STRUC-EMB: THE POTENTIAL OF STRUCTURE-AWARE ENCODING IN LANGUAGE EMBEDDINGS

## ABSTRACT

Text embeddings from Large Language Models (LLMs) have become founda-
tional for numerous applications. However, these models typically operate on raw
text, overlooking the rich structural information, such as hyperlinks or citations,
that provides crucial context in many real-world datasets. This paper introduces
and systematically evaluates a new paradigm for generating structure-aware text
embeddings by integrating these structural relations directly into the LLM's inter-
nal encoding process, rather than relying on traditional post-hoc aggregation. We
investigate two primary in-process methods: sequential concatenation and paral-
lel caching. Through extensive zero-shot experiments across retrieval, clustering,
classification, and recommendation tasks, we demonstrate that our structure-aware
approaches consistently outperform both text-only and post-hoc baselines. Our
analysis reveals critical trade-offs: sequential concatenation excels with noisy,
moderate-length contexts, while parallel caching scales more effectively to long,
high-signal contexts but is more susceptible to distractors. To address the chal-
lenge of noisy structural data, we also introduce and validate two effective tech-
niques: Context Distillation and Semantic Balancing. This work provides the
first comprehensive analysis of in-process structure-aware encoding, offering a
blueprint for building more powerful and contextually aware embedding models.

## 1 INTRODUCTION

Text embeddings form a critical foundation for numerous downstream applications, including infor-
mation retrieval, clustering, reranking, and recommendation (Lewis et al., 2020; Karpukhin et al.,
2020; Heffernan et al., 2022; Zhao et al., 2023). Progress from early BERT-style encoders (Reimers
& Gurevych, 2019; Liu et al., 2019) to recent large language model (LLM)–based embeddings
models (Zhang et al., 2025; Lee et al.; Wang et al., 2024b; BehnamGhader et al., 2024), has driven
state-of-the-art performance on benchmarks such as MTEB (Muennighoff et al., 2022; Enevoldsen
et al., 2025), underscoring the model capability to produce general-purpose vector representations.

Many real-world datasets contain not only raw text but also rich *structural information* that offers
complementary context for a variety of tasks. For instance, Wikipedia paragraphs are interconnected
by hyperlinks that serve as additional evidence for retrieval (Trivedi et al., 2022; Yang et al., 2018;
Ho et al., 2020), while e-commerce product recommendations can be guided by co-purchase and
co-view graphs (Ni et al., 2019). Such structural signals have recently been leveraged to enhance
LLMs in many generative tasks. Retrieval-augmented generation (RAG) systems, for example,
exploit document relations to improve multi-hop question answering (Wang et al., 2024d; Li et al.;
Han et al., 2025), and scientific summarization models benefit from a paper's internal structure
and citation network (Wang et al., 2024c; Luo et al., 2023; Edge et al., 2024; Zhao et al., 2024a).
However, the effective integration of this structural information into LLM-based embedding models
remains largely unexplored. Such a gap motivates this study for the following question: How can we
integrate structural information with the internal knowledge of powerful LLM encoders to improve
text embedding quality?

Previous work has integrated structural information with text embeddings, but typically as a post-hoc
step. In this paradigm, an LLM first encodes the target text, and then the embeddings of structurally
relevant documents are combined using simple aggregation functions like averaging or concatena-
tion (Bendada et al., 2023; Hou et al., 2022; Okura et al., 2017). However, such simple operations

are often poor proxies for even a simple summary, let alone the complex semantic synthesis required by many tasks. More sophisticated approaches employ trainable modules to learn task-specific aggregation functions, as seen in multi-document RAG and tasks involving text-attributed graphs or tabular data (Chen et al., 2024a; Abdaoui & Dutta, 2023; Wang et al., b; de Jong et al., 2023; Izacard et al., 2023). While more powerful, these methods have their own challenge: a significant dependency on task-specific labeled data for training and alignment (Zhao et al., 2024b; Li et al., 2024b).

In contrast to post-hoc approaches, our study explores a new paradigm for generating structure-aware embeddings by directly incorporating structural relations into the LLM encoding process itself. A straightforward method is sequential concatenation (later named **Struc-Emb-Seq**), where the target and its related segments are merged into a single sequence and encoded jointly. This mirrors the standard pipeline for context-augmented generation, where retrieved documents are concatenated with a query before being fed into an LLM; however, the objective here is to produce an embedding rather than a textual answer (Lewis et al., 2020; Wang et al., 2024e). This Struc-Emb-Seq approach aligns well with an LLM's pre-training on sequential text, preserving all token-level dependencies between the target and its context. Nevertheless, this method faces challenges as sequence length increases, including prohibitive computational costs, rapid context window consumption, and positional biases arising from the ordering of segments (Liu et al., 2023; Wu et al.; Lu et al., 2022). Crucially, long contexts risk diluting key information, a phenomenon known as the "needle-in-a-haystack" effect (Hsieh et al., 2024; Wang et al., 2025b; Li et al., 2024a).

A complementary approach, which we name **Struc-Emb-Par**, involves caching Key-Value (KV) states. In this method, each related segment is encoded individually, and its KV states are cached as condensed representations. The target segment then attends to these cached KVs without computing attention between the context segments themselves. The primary advantage is computational efficiency: segment KVs can be pre-computed and reused, and the online attention computation for any given target has only linear complexity. With proper allocation of positional encodings (PE), this method can also mitigate positional bias and context window limitations. However, it suffers from two major drawbacks. First, it fails to model interactions between the context segments. Second, it introduces a distribution shift, as attending to parallel, individually encoded KVs mismatches the LLM's sequential pre-training, a phenomenon shown to cause non-trivial performance degradation in generative tasks (Yang et al.; Ma et al., 2024). Some more detailed comparison between this approach and the previous parallel encoding for RAG is made in Sec. 2.

However, real-world structural relationships can be noisy. Consequently, incorporating structurally related segments may introduce irrelevant information and degrade the final embedding quality. To mitigate this issue and balance the influence of the target text against its structural context, we explore two techniques. The first is **Context Distillation**, which inserts an instruction prompt to guide the LLM to internally summarize and distill the related segments during the encoding process. This is analogous to methods that summarize conversational history in generative LLMs . The key difference, however, is that for an encoder model, this distillation occurs implicitly within the model's internal states rather than by explicitly generating summary tokens. The second technique is **Semantic Balancing**. This method combines the structure-aware embedding (derived from the target and its related segments) with the standalone embedding of the target segment. Properly balancing these two representations allows for explicit control over the amount of information contributed by the structural context versus the original target content.

To the best of our knowledge, this is the first work to study structure-aware text embedding by integrating structural information directly into the LLM's internal encoding process. We evaluate different methods in a zero-shot setting. We conduct extensive experiments on retrieval (Yang et al., 2018; Trivedi et al., 2022), clustering (Muennighoff et al., 2022), product recommendation (Wu et al., 2024; Ni et al., 2019), and citation classification (Sen et al., 2008) and observe several key findings: 1) Incorporating structural information consistently improves performance over text-only embeddings, with the largest gains in tasks where text alone is insufficient, such as multi-hop question answering; 2) The in-process approach of encoding segments jointly yields better embeddings than traditional post-hoc aggregation, particularly when the structural context is large and noisy; 3) Among the Struc-Emb variants, sequential concatenation (Struc-Emb-Seq) excels on noisy, moderate-length inputs but is sensitive to order and degrades on long texts, while parallel caching (Struc-Emb-Par) variants scale robustly to long contexts with high-signal data but are more susceptible to distractors; 4) Both Context Distillation and Semantic Balancing are effective techniques for preserving the target's core semantics when encountering such noisy structural context.

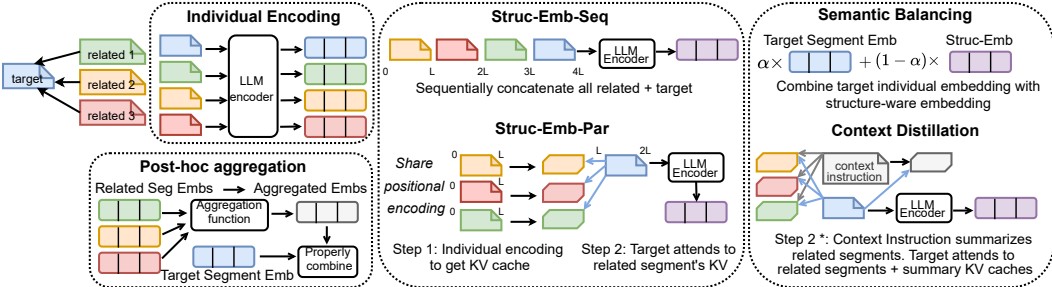

Figure 1: Given a target and its related segments, individual encoding and post-hoc aggregation serve as baselines that separate embedding and structure aggregation. Structure-aware encoding instead injects structural relations during encoding via Struc-Emb-Seq and Struc-Emb-Par, further enhanced by semantic balancing and context distillation for robust structural information utilization.

## 2 RELATED WORKS

**Text Embedding Models.** Recent text embedding models have shifted from traditional encoder-only architectures (Devlin et al., 2019; Reimers & Gurevych, 2019; Liu et al., 2019) to decoder-only LLMs, achieving state-of-the-art performance on benchmarks like MTEB (Muennighoff et al., 2022). Decoder-only generative LLMs (Jiang et al., 2023; Yang et al., 2025a) can be finetuned to become embedding models: Some of them explicitly remove or relax the causal mask to enable bidirectional context modeling (Meng et al., 2024; BehnamGhader et al., 2024; Muennighoff et al., 2024; Lee et al., 2024), while others retain the unidirectional causal attention of standard LLMs (Wang et al., 2024b; OpenAI, 2024; Zhang et al., 2025), where careful instruction tuning and large-scale synthetic training can overcome the limitations of left-to-right encoding.

**Modeling structured data with LLMs.** Prior work on structured data with LLMs has primarily targeted generative tasks rather than encoding. One line of research serializes structures into text (e.g., verbalization) for direct processing (Fatemi et al.; Perozzi et al., 2024). The second strategy projects graph features into the LLM's token embedding space (Kong et al.; Chen et al., 2024a; Wang et al., 2024a), but this often suffers from generalization challenges in aligning the structure and text modalities (Chen et al., 2024b; Li et al., 2024b). A recent work Graph-KV (Wang et al., 2025a) offers a distinct approach by modifying the LLM's attention mechanism for structured bias injection for solving generation tasks.

**Parallel Encoding in Generative Models.** Several recent works have proposed caching the KV states of multiple documents in parallel to reuse them for downstream generative tasks like RAG (Yang et al.; Ratner et al., 2023; Gim et al., 2024; Ma et al., 2024; Wang et al., 2025a; Yang et al., 2025b). The primary motivation for these methods was to improve computational efficiency. A known drawback is that this parallel processing introduces a distribution shift from the model's sequential pre-training, often necessitating further LLM fine-tuning to mitigate performance degradation. However, this technique has not been previously studied for the task of generating text embeddings, let alone for integrating real-world structural information as context. For graph-connected text segments, this can be viewed as a message passing process to compute text embeddings. We hypothesize that parallel caching may be *more promising* for embedding models, because embedding generation is inherently more robust to this distribution shift compared to autoregressive text generation, where errors can compound during token-by-token rollout. This makes parallel caching a viable and compelling strategy for our structure-aware embedding paradigm.

## 3 METHODOLOGY

In this section, we describe the structured text embedding task and baselines. We then explore structure-aware encoding by injecting structural information directly into the encoding process, with Struc-Emb-Seq and Struc-Emb-Par. Lastly, to improve robustness over contextual information, we explore context distillation over related texts and semantic balancing to preserve target semantics.

### 3.1 PRELIMINARIES AND BASELINES

**Preliminaries.** We study a setting where each text segment can be linked to structurally related segments through predefined relations. Formally, let $\mathcal{D} = \{u_1, u_2, \ldots, u_n\}$ be the collection of text

segments. For each $u_i$, its related set is $\mathcal{N}_i = \{v_j \mid (u_i, v_j) \in \mathcal{R}\}$, where $(u_i, v_j)$ denotes a directed relation. This forms a star-shaped structure with $u_i$ as the target and $\mathcal{N}_i$ as its related segments. The objective is to generate embeddings for all $u_i$'s while incorporating information from $\mathcal{N}_i$.

**Baselines.** We consider baselines using a pretrained embedding model without modifying its encoding process:

- *Individual embedding:* Encode $u_i$ independently with a pretrained embedding model. This represents the vanilla encoding process without leveraging related text segments.
- *Post-hoc aggregation:* Encode $u_i$ and its related segments $v_j \in \mathcal{N}_i$ independently into embeddings $h_i, h_j \in \mathbb{R}^d$. Aggregate embeddings from related segments into $h_i^{\text{agg}}$ while keeping $h_i$ unchanged. We consider (i) mean pooling, the uniform average of all related embeddings, and (ii) weighted pooling, where weights are softmax-normalized cosine similarities with $h_i$:

$$h_i^{\text{agg}} = \frac{1}{|\mathcal{N}_i|} \sum_{v_j \in \mathcal{N}_i} h_j, \qquad h_i^{\text{agg}} = \sum_{v_j \in \mathcal{N}_i} \frac{\exp(\cos(h_i, h_j))}{\sum_{v_k \in \mathcal{N}_i} \exp(\cos(h_i, h_k))} h_j.$$

As discussed in Sec. 3.3, $h_i^{\text{agg}}$ can later be interpolated with $h_i$.

Individual embedding ignores structural relations and post-hoc aggregation might miss low-level word or sentence level interactions and encounter other issues discussed in Sec. 1. Thus, we next investigate structure-aware methods that incorporate related segments directly during encoding.

## 3.2 STRUCTURE-AWARE ENCODING

In this subsection, we investigate two ways, either sequentially or in parallel, to encode structural information to generate structure-aware embeddings of target text segment.

**Struc-Emb-Seq** Given a target segment $u_i$ and its related segments $\{v_1, \ldots, v_n\}$, we concatenate them into a single sequence $[v_1, \ldots, v_n, u_i]$ and encode it with the pretrained embedding model.

The primary strength of Struc-Emb-Seq is that its sequential format matches the pretraining of LLM encoders, which are optimized to model semantics in continuous text. This alignment allows the model fully exploit its inherent capabilities. Encoding the target and all related segments in a single sequence enables self-attention to capture fine-grained, long-range dependencies, giving the target direct access to the entire context.

Computationally, full self-attention over the concatenated sequence has $\mathcal{O}(n^2 L^2)$ complexity, where $L$ is the expected segment length. Since each segment uses distinct positional encodings (PEs), the context window grows linearly with the number of related texts, limiting how many text segments can be included, and the embeddings are sensitive to concatenation order. Semantically, mixing useful and irrelevant content may bury important signals, similar to a needle-in-a-haystack problem observed in long-context generative models (Liu et al., 2023; Hsieh et al., 2024).

**Struc-Emb-Par.** This method is an alternative way to encode the structurlly related segments with the target. As illustrated in Fig. 1, first, each related (contextual) segment is encoded independently, producing parallel cached KV pairs $(K_{r,1}, V_{r,1}), \ldots, (K_{r,n}, V_{r,n})$ without any imposed ordering across segments. In the second stage, the target segment $u_t$ is encoded, and its queries $Q_t$ attend both to its own keys and values $(K_t, V_t)$ and to the cached KVs from the contextual segments:

$$\text{Attention}\big(Q_t, ; [K_t; K_{r,1}; \ldots; K_{r,n}], ; [V_t; V_{r,1}; \ldots; V_{r,n}]\big).$$

There are no attentions between two contextual segments. So, the attention structure is sparse according to the structural relations. The attention aggregation to compute the target segment embedding works as weighted message passing from the context's KV caches, analogous to graph attention networks (Velickovic et al., 2017). It offers both efficiency and semantic benefits: The greatest advantage is that contextual caches can be precomputed once and reused across targets, avoiding costly re-encoding whenever the set or order of segments changes. In addition, the sparse attention reduces the computation from $\mathcal{O}(n^2 L^2)$ in Struc-Emb-Seq to $\mathcal{O}(nL^2)$. Moreover, by imposing that all contextual segments share PE range $[0, L]$ and the target uses $[L, 2L]$, keeping context window usage fixed at $2L$ regardless of the number of segments. On the semantic side, treating contextual segment KV caches as parallel and unordered avoids positional biases that Struc-Emb-Seq may suffer from. One drawback of Struc-Emb-Par is that cross-contextual-segment interactions are not modeled.

Another key issue is that Struc-Emb-Par diverges from the sequential nature of LLM encoders, which are pretrained on continuous text with attention over one ordered sequence. In Struc-Emb-Par, the target attends to cached KVs from individually encoded segments. This mismatch may shift the attention distributions.

## 3.3 CONTEXT DISTILLATION AND SEMANTIC BALANCING

Indeed, incorporating structural relations may also introduce noisy and irrelevant information that overwhelms and dilutes the target's original semantics. To mitigate this issue, we propose two techniques: **Context Distillation** and **Semantic Balancing**. Context Distillation aims to summarize the most relevant context in latent space to guide target encoding. Semantic Balancing fuses the structure-aware embedding with the original target embedding, controlling the degree of contextual information without overriding the target's semantics.

**Context Distillation** We extend Struc-Emb-Par with a distillation step as shown in Fig. 1, named Struc-Emb-Par-Distill. After producing cached KVs $\{(K_{r,j}, V_{r,j})\}$ as in Struc-Emb-Par, a context instruction is injected that attends over these caches to form a distilled cache $(K_c, V_c)$, which summarizes all contextual segments in the latent space. When encoding the target $u_t$, its queries $Q_t$ attend to its own states, the caches of contextual segments, and the distilled cache:

$$\text{Attention}\big(Q_t, ; [K_t; K_c; K_{r,1}; \ldots; K_{r,n}], ; [V_t; V_c; V_{r,1}; \ldots; V_{r,n}]\big).$$

The context instruction takes the form: "Summarize [related paragraphs] into a contextual representation that captures [key shared concepts]. Use this distilled context, along with the original paragraphs as supporting evidence, when encoding the following [target segment type] for [task objective]: ". Complete instructions for each dataset are provided in A.3.

Struc-Emb-Par-Distill enhances Struc-Emb-Par by generating a distilled KV state that summarizes salient information, such as shared entities and core themes, from all related segments. The distilled summary determines the overall context, while the original caches preserve fine-grained details, so combining both improves robustness to noise without losing details. Importantly, this summarization occurs entirely *inside* LLMs, which is different from the summary of conversational history in generative LLMs. In contrast, our preliminary attempts to add similar context instructions to Struc-Emb-Seq had limited effect, as it relies more on explicit text summaries.

**Semantic Balancing: Interpolation with Individual Embedding.** To preserve target semantics while incorporating context, we also consider interpolating between the original target embedding and embedding containing structural information:

$$h_i = (1 - \alpha)\, h_i^{\text{individual}} + \alpha\, h_i^{\text{struct}},$$

where $h_i^{\text{struct}}$ can be structure-aware embedding produced from Struc-Emb-Seq, Struc-Emb-Par, Struc-Emb-Par-Distill, or aggregated embedding $h_i^{\text{agg}}$ from the post-hoc aggregation in Sec. 3.1. The coefficient $\alpha \in [0, 1]$ controls the balance between contextual and target information.

## 4 EXPERIMENTS AND ANALYSIS

### 4.1 BACKBONES AND BASELINES

The baselines are *individual encoding* and *post-hoc aggregation* as described in Sec. 3.1. For structure-aware methods, we evaluate *Struc-Emb-Seq*, *Struc-Emb-Par*, *Struc-Emb-Par-Distill*, and their corresponding performance under semantic balancing when interpolating with individual embedding. We select the interpolation coefficient $\alpha$ by grid search over $[0, 1]$ directly on the test set, aiming to measure the upper bound of both post-hoc aggregation and our structure-aware methods. The search range and the optimal $\alpha$ for each dataset are reported in A.5.3. All baselines and Struc-Emb variants are built on pretrained LLM embedding models: Qwen3-Embedding (0.6B, 4B) and E5-Mistral-Instruct-7B. We select these models for their strong MTEB performance and because their unidirectional causal attention lets the target segment attend to preceding related segments by injecting related segment KVs as past KVs. The bidirectional encoders may require modified mechanism as discussed in A.2. Main texts include Qwen3 results with E5-Mistral in A.5.2.

## 4.2 DATASETS AND TASKS

To evaluate the impact of structure-aware embeddings, we consider five downstream tasks spanning nine datasets. Below, we briefly summarize each dataset along with its task and associated structural information. Full dataset details are provided in A.1.

**Multi-hop QA Retrieval.** We first evaluate on the *MuSiQue* dataset (Trivedi et al., 2022). Each question is paired with 20 candidate Wikipedia paragraphs, of which 2–4 are ground-truth evidence. Structural relations are defined at the page level: two paragraphs are linked if their source pages are connected by a hyperlink. Thus, within each question, any two candidates from linked pages are treated as related. Further, for each candidate paragraph, we add additional related paragraphs from pages hyperlinked by its source page but not already included among the 20 candidates. Among these linked pages, we select the most relevant ones using three criteria: (i) highest PageRank score in the Wikipedia link graph, (ii) largest hyperlink degree, and (iii) closest semantic similarity to the candidate paragraph. To discover the impact of context number, we include a total of 5 or 10 linked paragraphs for every candidate. Performance is evaluated under two settings: (1) ranking paragraphs within each question's 20 candidates, and (2) retrieval from the global pool of all candidate paragraphs across questions, both using nDCG@10 and Recall@k as metrics.

We also use *HotpotQA* (Yang et al., 2018) from MTEB and BEIR (Thakur et al.). Each question is associated with two supporting Wikipedia documents. We treat these paired documents as having a structural relation and evaluate retrieval with nDCG@10 and Recall@k.

**Citation Network Paper Classification.**(Yang et al., 2016) We evaluate on *Cora*, *Citeseer*, and *Pubmed*, where the task is to classify papers into topics using titles and abstracts as text and citation links as structure. Following Wang et al. (a), we construct class embeddings by sampling $20\times$ (number of classes) papers, labeling them with GPT-4o, and averaging their embeddings. Papers are assigned to the nearest class embedding by cosine similarity, evaluated with accuracy and macro F1.

**E-commerce Product Classification.** (Ni et al., 2019) The *Books-History* dataset contains Amazon "History" books, using titles and descriptions as text and co-purchase/co-view links as structure. The task is to classify each book into 12 categories. The *Sports-Fitness* dataset consists of Amazon fitness products, using titles as text with the same structural relations. The task is to classify each item into 13 categories. Evaluation follows the citation network classification setup above.

**E-commerce Product Recommendation.** (Wu et al., 2024) We use the *STaRK-Amazon* dataset, which includes four entity types (product, brand, color, category) and five relations (also-bought, also-viewed, has-brand, has-color, has-category). Products have rich text attributes such as descriptions, reviews, and Q&A, while other entities use names or titles. The task is to recommend products from either human-generated or synthetic queries, evaluated with Hit@k, Recall@k and MRR following the original paper. Since the synthetic queries only contains up to 20 answers, so we only evaluate on Recall@20, while human queries may contain answers up to 88.

**Stack Exchange Post Clustering.** We use the *StackExchangeClustering* dataset from MTEB and derive structural information from metadata in the original StackExchange corpus[1] . The task is to cluster post titles, where two posts are linked if they share the same set of tags. Evaluation follows MTEB using V-measure (Rosenberg & Hirschberg, 2007).

## 4.3 FINDING DISCUSSIONS

**RQ1: Does structure information provide additional benefit over the individual embeddings?**

This question examines under what conditions incorporating structural information yields improvements beyond individual embeddings. Our analysis across datasets suggests the following pattern.

*1). Structural information benefits more when texts alone are insufficient or under-represented.*

Across datasets, embeddings including structural information yield the largest gains when the textual signals are weak. For instance, multi-hop QA requires combining evidence across documents, and critical information may not appear in a single document but is captured in linked entities, which

---

[1]https://huggingface.co/datasets/flax-sentence-embeddings/
stackexchange_title_body_jsonl

Table 1: Results for MuSiQue datasets. "w. balancing" denotes semantic balancing. For each setting (w. and w/balancing), the best method is in **bold** and the second best is underlined. Ret. reports nDCG@10 over all candidates, while Rank. reports nDCG@10 over candidates per question.

| | METHOD | TOP-5 NEIGHBORS | | | | | | TOP-10 NEIGHBORS | | | | | |
|---|---|---|---|---|---|---|---|---|---|---|---|---|---|
| | | DEGREE | | SEMANTIC | | PAGERANK | | DEGREE | | SEMANTIC | | PAGERANK | |
| | | RET. | RANK. | RET. | RANK. | RET. | RANK. | RET. | RANK. | RET. | RANK. | RET. | RANK. |
| | QWEN3-EMBEDDING 0.6B | | | | | | | | | | | | |
| W/ BALANCING | INDIVIDUAL | 54.10 | 78.62 | 54.10 | 78.62 | 54.10 | 78.62 | 54.10 | 78.62 | 54.10 | 78.62 | 54.10 | 78.62 |
| | STRUC-EMB-SEQ | 61.91 | 83.66 | 63.46 | 82.96 | 61.81 | 83.07 | 51.29 | 78.06 | 49.04 | 78.39 | 50.71 | 78.33 |
| | STRUC-EMB-PAR | 54.77 | 80.85 | 57.89 | 81.48 | 54.61 | 81.08 | 35.53 | 71.76 | 39.17 | 74.04 | 35.31 | 72.14 |
| | STRUC-EMB-PAR-DISTILL | 61.84 | 83.65 | 60.76 | 83.51 | 60.89 | 83.04 | 40.11 | 74.51 | 42.42 | 75.79 | 40.40 | 74.54 |
| W. BALANCING | MEAN NEIGHBOR | 64.80 | 85.98 | 63.83 | 85.24 | 64.83 | 86.03 | 59.24 | 82.34 | 59.42 | 82.62 | 59.15 | 82.45 |
| | WEIGHTED MEAN NEIGHBOR | 67.72 | 87.42 | 62.59 | 84.48 | 67.32 | 87.23 | 60.06 | 82.92 | 58.89 | 82.04 | 59.99 | 83.07 |
| | STRUC-EMB-SEQ | 73.71 | 88.49 | 71.27 | 86.91 | 73.99 | 88.40 | 69.53 | 86.97 | 62.69 | 84.88 | 69.58 | 86.84 |
| | STRUC-EMB-PAR | 66.49 | 85.84 | 65.24 | 85.01 | 66.24 | 85.79 | 61.00 | 82.49 | 57.99 | 81.28 | 60.90 | 82.43 |
| | STRUC-EMB-PAR-DISTILL | 66.91 | 85.99 | 65.47 | 85.39 | 67.25 | 86.13 | 61.28 | 82.82 | 58.09 | 81.53 | 61.32 | 82.68 |
| | QWEN3-EMBEDDING 4B | | | | | | | | | | | | |
| W/ BALANCING | INDIVIDUAL | 59.45 | 83.49 | 59.45 | 83.49 | 59.45 | 83.49 | 59.45 | 83.49 | 59.45 | 83.49 | 59.45 | 83.49 |
| | STRUC-EMB-SEQ | 70.78 | 87.31 | 74.16 | 88.04 | 70.51 | 87.01 | 61.22 | 83.32 | 61.45 | 84.3 | 61.87 | 83.79 |
| | STRUC-EMB-PAR | 62.97 | 85.79 | 68.48 | 87.13 | 62.55 | 85.44 | 40.54 | 75.70 | 48.98 | 80.21 | 40.98 | 75.98 |
| | STRUC-EMB-PAR-DISTILL | 63.38 | 86.48 | 68.01 | 87.49 | 63.78 | 86.34 | 42.77 | 76.60 | 53.12 | 81.80 | 43.43 | 76.80 |
| W. BALANCING | MEAN NEIGHBOR | 71.72 | 89.40 | 70.04 | 88.55 | 71.22 | 89.22 | 65.00 | 86.43 | 65.12 | 86.30 | 64.74 | 86.43 |
| | WEIGHTED MEAN NEIGHBOR | 73.28 | 90.09 | 68.37 | 87.83 | 73.25 | 89.99 | 65.95 | 86.70 | 64.56 | 85.71 | 65.60 | 86.79 |
| | STRUC-EMB-SEQ | 79.29 | 91.60 | 78.65 | 91.19 | 79.39 | 91.91 | 76.15 | 90.55 | 69.98 | 88.37 | 76.16 | 90.44 |
| | STRUC-EMB-PAR | 73.03 | 89.48 | 72.20 | 89.34 | 72.56 | 89.43 | 65.33 | 86.18 | 63.97 | 85.48 | 65.38 | 86.66 |
| | STRUC-EMB-PAR-DISTILL | 71.91 | 89.23 | 71.69 | 89.36 | 71.83 | 89.35 | 64.78 | 85.82 | 65.47 | 85.72 | 64.89 | 85.95 |

Table 2: Results for networked object classification. "w. balancing" denotes semantic balancing. For each setting (w. and w/balancing), the best method is in **bold** and the second best is underlined.

| | METHOD | CORA | | CITESEER | | PUBMED | | BOOKHIS | | SPORTSFIT | |
|---|---|---|---|---|---|---|---|---|---|---|---|
| | | ACC | F1 | ACC | F1 | ACC | F1 | ACC | F1 | ACC | F1 |
| | QWEN3-EMBEDDING 0.6B | | | | | | | | | | |
| W/ BALANCING | INDIVIDUAL | 73.06 | 68.11 | 66.48 | 61.72 | 81.24 | 81.14 | 60.58 | 26.09 | 56.66 | 45.01 |
| | STRUC-EMB-SEQ | 70.48 | 66.52 | 68.67 | 64.07 | 80.68 | 80.12 | 60.11 | 26.04 | 67.67 | 53.14 |
| | STRUC-EMB-PAR | 72.69 | 68.73 | 68.98 | 64.29 | 80.6 | 80.16 | 60.27 | 26.00 | 67.96 | 55.09 |
| | STRUC-EMB-PAR-DISTILL | 72.14 | 68.13 | 69.52 | 64.64 | 81.29 | 80.83 | 60.06 | 25.62 | 66.70 | 54.44 |
| W. BALANCING | MEAN NEIGHBOR | 75.09 | 71.02 | 69.13 | 64.33 | 81.87 | 81.76 | 61.09 | 26.58 | 67.33 | 53.13 |
| | WEIGHTED MEAN NEIGHBOR | 75.09 | 71.10 | 68.98 | 64.15 | 81.92 | 81.82 | 61.11 | 26.57 | 67.14 | 52.94 |
| | STRUC-EMB-SEQ | 74.91 | 70.74 | 69.25 | 64.57 | 83.11 | 82.82 | 61.05 | 27.22 | 67.76 | 53.58 |
| | STRUC-EMB-PAR | 75.09 | 70.96 | 69.06 | 64.34 | 82.71 | 82.48 | 61.17 | 27.21 | 67.96 | 55.10 |
| | STRUC-EMB-PAR-DISTILL | 75.09 | 70.75 | 69.60 | 64.75 | 82.89 | 82.55 | 61.51 | 27.69 | 66.70 | 54.48 |
| | QWEN3-EMBEDDING 4B | | | | | | | | | | |
| W/ BALANCING | INDIVIDUAL | 74.54 | 70.40 | 68.28 | 63.79 | 84.84 | 81.08 | 62.38 | 26.82 | 61.47 | 51.64 |
| | STRUC-EMB-SEQ | 74.17 | 70.76 | 68.16 | 63.50 | 82.35 | 81.40 | 61.11 | 25.72 | 70.93 | 58.85 |
| | STRUC-EMB-PAR | 74.17 | 70.60 | 69.10 | 64.43 | 82.15 | 81.19 | 61.29 | 26.86 | 71.15 | 61.38 |
| | STRUC-EMB-PAR-DISTILL | 73.80 | 70.06 | 69.29 | 64.54 | 81.82 | 81.02 | 61.38 | 26.25 | 71.85 | 61.87 |
| W. BALANCING | MEAN NEIGHBOR | 76.01 | 72.18 | 70.23 | 65.61 | 85.45 | 84.88 | 63.00 | 27.76 | 69.61 | 58.97 |
| | WEIGHTED MEAN NEIGHBOR | 75.83 | 72.05 | 70.19 | 65.53 | 85.47 | 84.90 | 62.95 | 27.77 | 69.55 | 58.96 |
| | STRUC-EMB-SEQ | 76.38 | 73.04 | 69.56 | 64.93 | 85.80 | 85.17 | 63.10 | 28.64 | 70.93 | 59.36 |
| | STRUC-EMB-PAR | 76.20 | 72.41 | 69.72 | 65.01 | 85.57 | 84.93 | 63.06 | 28.35 | 71.15 | 61.38 |
| | STRUC-EMB-PAR-DISTILL | 76.38 | 72.73 | 69.95 | 65.33 | 85.29 | 84.69 | 63.35 | 28.26 | 71.85 | 61.92 |

clearly surpass text-only embeddings as seen in MuSiQue and HotpotQA. Similarly, in SportsFit dataset, short and ambiguous titles benefit from co-purchase and co-view relations that supply missing context. Intermediate cases include StackExchange and STaRK-Amazon, where related segments provide moderate but useful additional information as discussed in (Wu et al., 2024). By contrast, in citation networks and BookHis, structural information yields limited benefit, as titles, abstracts, and descriptions already contain sufficient cues for category classification. The prior work (Wang et al., a) similarly finds that GPT-4o with text-only inputs achieves strong results. In these stronger-text settings, structural information becomes more useful with smaller models such as the 0.6B encoder, whose weaker text representations benefit more from external context.

*2). The effectiveness of structural information also depends on the number and quality of neighbors.*

In practice, the gain from structural information also depends on the quality of related segments and is maximized with accurate structural links. In HotpotQA, where neighbors are gold-standard evidence, incorporating this "perfect" structural context yields substantial performance gains. In contrast, noisy contexts may dimish effectiveness: in MuSiQue, using the top-5 related segments outperforms the top-10, as adding more irrelevant Wikipedia pages dilutes the useful context. The type of neighbors also matters: semantically similar but contextually irrelevant nodes can be particularly distracting, leading to lower performance compared to the other two selection methods.

*3). Struc-Embs alone can underperform individual embeddings when neighbor noise or distributional shifts dominate, underscoring the need to for context distillation and semantic balancing.*

As analyzed in Sec. 3.2, Struc-Emb-Seq suffers from mixing noise with signals and context-window limits, while parallel variants face distributional shifts from processing parallel caches simultaneously. Empirically, Struc-Emb-Seq drops noticeably from 5 to 10 related segments in MuSiQue and further degrades on long texts in Stark. The parallel variants decay fast as the number of irrelevant

Table 3: Results for STaRK-Amazon dataset. "w. balancing" denotes semantic balancing. For each setting (w. and w/balancing), the best method is in **bold** and the second best is underlined.

| | METHOD | HUMAN-GENERATED | | | | | | | SYNTHETIC | | | |
|---|---|---|---|---|---|---|---|---|---|---|---|---|
| | | HIT@1 | HIT@5 | R@20 | R@30 | R@50 | R@90 | MRR | HIT@1 | HIT@5 | R@20 | MRR |
| | QWEN3-EMBEDDING 0.6B | | | | | | | | | | | |
| W/ BALANCING | INDIVIDUAL | **79.01** | 90.12 | **73.78** | **80.1** | 87.90 | **95.71** | **83.39** | **67.36** | 87.15 | **78.23** | **76.04** |
| | STRUC-EMB-SEQ | 70.37 | 83.95 | 62.14 | 70.59 | 80.23 | 87.65 | 76.62 | 60.54 | 80.21 | 67.94 | 69.40 |
| | STRUC-EMB-PAR | 74.07 | 90.12 | 69.89 | 79.65 | **89.64** | 94.99 | 81.52 | 61.57 | 82.40 | 72.96 | 71.07 |
| | STRUC-EMB-PAR-DISTILL | 74.07 | **91.36** | 71.26 | 79.29 | 89.07 | 94.62 | 81.81 | 62.67 | 84.29 | 74.17 | 72.19 |
| W. BALANCING | MEAN NEIGHBOR | 80.25 | 91.59 | 73.83 | 80.99 | 90.65 | 96.00 | 85.83 | 66.77 | 87.76 | 78.89 | 76.11 |
| | WEIGHTED MEAN NEIGHBOR | **81.48** | 92.59 | 73.84 | 80.96 | 90.78 | 95.93 | **86.26** | 68.03 | 89.22 | 79.91 | 78.33 |
| | STRUC-EMB-SEQ | 80.25 | **93.83** | 75.27 | 82.02 | 91.77 | 96.79 | 85.58 | **69.73** | 89.22 | 79.91 | 78.33 |
| | STRUC-EMB-PAR | 80.25 | 92.59 | 75.16 | **83.25** | **91.92** | **96.97** | 85.14 | 69.00 | **89.34** | 80.02 | 77.91 |
| | STRUC-EMB-PAR-DISTILL | 80.25 | **93.83** | 75.29 | 81.86 | 91.50 | 96.94 | 85.97 | 69.18 | 89.16 | **80.18** | 78.04 |
| | QWEN3-EMBEDDING 4B | | | | | | | | | | | |
| W/ BALANCING | INDIVIDUAL | **83.95** | 93.83 | 76.64 | 83.74 | 91.59 | 96.41 | **89.07** | **71.07** | 88.25 | 81.55 | **78.81** |
| | STRUC-EMB-SEQ | 77.78 | 90.12 | 64.84 | 74.20 | 83.71 | 89.37 | 83.42 | 61.75 | 80.58 | 69.86 | 70.28 |
| | STRUC-EMB-PAR | 80.25 | **96.30** | 75.64 | 83.50 | 91.78 | **97.00** | 85.81 | 64.31 | 85.93 | 77.71 | 73.90 |
| | STRUC-EMB-PAR-DISTILL | 81.48 | 93.83 | 75.19 | 83.68 | 90.92 | 96.73 | 87.12 | 64.49 | 85.93 | 76.95 | 74.13 |
| W. BALANCING | MEAN NEIGHBOR | **88.89** | 96.30 | 77.11 | 83.90 | 91.99 | 97.59 | 91.58 | 71.92 | 90.13 | 82.14 | 79.49 |
| | WEIGHTED MEAN NEIGHBOR | 87.65 | 96.30 | 77.11 | 83.90 | 91.89 | 97.59 | 91.44 | 71.80 | 89.89 | 82.11 | 79.44 |
| | STRUC-EMB-SEQ | **88.89** | 96.30 | 77.51 | 84.19 | 92.16 | 98.34 | 92.40 | 73.75 | 90.07 | 82.73 | 81.05 |
| | STRUC-EMB-PAR | **88.89** | 96.30 | 77.27 | 83.87 | 92.44 | 98.32 | 92.15 | 73.26 | 89.89 | 83.24 | 80.78 |
| | STRUC-EMB-PAR-DISTILL | **88.89** | 96.30 | 77.40 | 84.08 | **93.07** | **98.36** | 92.34 | 73.57 | 90.50 | 83.31 | 80.90 |

Table 4: Results for StackExchange clustering (Clust.) evaluated with V-measure and HotpotQA dataset. "w. balancing" denotes semantic balancing. For each setting (w. and w/balancing), the best method is in **bold** and the second best is underlined.

| | METHOD | QWEN3-EMBEDDING 0.6B | | | | QWEN3-EMBEDDING 4B | | | |
|---|---|---|---|---|---|---|---|---|---|
| | | CLUST. | HotpotQA | | | CLUST. | HotpotQA | | |
| | | | NDCG@10 | R@5 | R@10 | | NDCG@10 | R@5 | R@10 |
| W/ BALANCING | INDIVIDUAL | 61.33 | 83.99 | 81.18 | 84.87 | 67.89 | 89.54 | 87.85 | 91.13 |
| | STRUC-EMB-SEQ | 59.60 | 96.26 | 97.12 | 98.14 | 64.66 | 97.65 | 98.31 | 99.01 |
| | STRUC-EMB-PAR | 64.98 | 96.34 | 97.02 | 98.10 | 64.95 | 97.73 | 98.25 | 98.95 |
| | STRUC-EMB-PAR-DISTILL | 64.72 | **97.47** | **98.10** | **98.77** | 75.21 | **98.32** | **98.80** | **99.29** |
| W/ BALANCING | MEAN NEIGHBOR | **70.02** | 97.18 | 100.00 | 100.00 | 71.52 | 98.30 | 100.00 | 100.00 |
| | WEIGHTED MEAN NEIGHBOR | 69.72 | 97.17 | 100.00 | 100.00 | 71.54 | 98.31 | 100.00 | 100.00 |
| | STRUC-EMB-SEQ | 69.40 | 97.20 | 100.00 | 100.00 | 73.53 | 98.45 | 100.00 | 100.00 |
| | STRUC-EMB-PAR | 69.74 | 97.24 | 100.00 | 100.00 | 71.35 | **98.52** | 100.00 | 100.00 |
| | STRUC-EMB-PAR-DISTILL | 69.18 | **97.47** | 100.00 | 100.00 | 75.78 | 98.50 | 100.00 | 100.00 |

segments grows in MuSiQue. These results validate our analysis and highlight the importance of context distillation and balancing structural context with target semantics (will discuss in RQ 2.2).

**RQ2: What is the most effective way to incorporate structural information into embeddings?**

● **2.1: Comparing Struc-Emb-Seq and Struc-Emb-Par variants**

*1). Sequential excels on noisy, moderate-length inputs but degrades on long texts. Parallel scales robustly with long context, excels at high-signal context but is more susceptible to distractors.*

Struc-Emb-Seq performs best when noisy segments are present but the sequence length is moderate, as in MuSiQue. In this setting, it leverages the model's implicit ability to summarize and filter sequential context without yet suffering from long-context degradation. Its weakness emerges on longer texts and increasing related segments, such as STaRK-Amazon and Bookhis, where context-window limits and long-context issues—positional bias, lost-in-the-middle, and needle-in-a-haystack effects—reduce effectiveness. In contrast, Struc-Emb-Par variants struggle to filter and weight parallel inputs when many distractors are present, as in MuSiQue with 10 related segments. However, when noise is limited, they sometimes outperform sequential, as in citation networks and clustering, and achieve strong results when relations are highly informative, such as in HotpotQA and Sports-Fitness. Lastly, Struc-Emb-Par variants are also more robust to long texts, since individually encoded caches with shared PE stabilizes window usage and avoids positional bias.

We further compare Struc-Emb-Seq and Struc-Emb-Par regarding computation time, performance sensitivity to order of related segments and robustness to context length in App. A.4. The Struc-Emb-Par variants presents similar scale of encoding time with individual encoding given cached related KVs, while Struc-Emb-Seq incurs considerably higher costs. Struc-Emb-Seq is also sensitive to the order in which related segments are concatenated. In MuSiQue dataset, the default ordering places linked paragraphs from the same question first, implicitly ranking by relevance, thus yields much higher performance than random permutation. In addition, Struc-Emb-Seq decays faster than Struc-Emb-Par as text length grows in both target and related segments, and eventually underperforms once the sequence exceeds the context window limit set.

● **2.2: The impacts from Context Distillation and Semantic Balancing**

As pointed out in RQ1 finding 3), our motivation of context distillation and semantic balancing is valid through empirical results. Next, we investigate in what conditions they benefit the most.

*1). Context distillation helps under when presenting noisy segments, while Struc-Emb-Par can be competitive under small number of informative segments.*

When compared individually, Struc-Emb-Par-Distill consistently outperforms Struc-Emb-Par in the presence of noisy related segments, with notable gains under strong noise as in MuSiQue and, in most cases, smaller gains when the irrelevant segments are less distracting, as in STaRK-Amazon. In MuSiQue particular, Struc-Emb-Par-Distill provides significant benefits under top-10 related segments and exhibits lower variance across different neighbor selection strategies. In contrast, given less related segments with minimal noise, such as in networked object classification, Struc-Emb-Par sometimes perform better, especially when the context is highly informative as in Sports-Fitness.

With semantic balancing, the gain from context distillation diminishes because interpolation calibrates toward target semantics. This effect depends on the task: for high-level tasks (e.g., classification, clustering), the distilled summary in Struc-Emb-Par-Distill remains beneficial, whereas for detail-oriented retrieval tasks (e.g., MuSiQue, STaRK-Amazon), raw segments are more effective.

*2). Semantic balancing is essential to preserve target information, and the interpolation coefficient $\alpha$ reflects task reliance on structural information and the quality of the context.*

Combining to individual embedding always improves performance over all pipelines, proving that structure-ware encodings contain issue in diluting target semantics which require some explicit balancing. And preserving target semantics is always critical.

The optimal $\alpha$ values (in A.5.3.) from our oracle exploration reflect the effectiveness of structural information, with lower $\alpha$ indicating greater reliance on structural signals. We observe three consistent patterns. 1). Datasets that benefit more from structure show lower $\alpha$, with the smallest values in HotpotQA, Sports-Fitness, and MuSiQue, followed by clustering tasks, and higher values in STaRK-Amazon, citation networks, Book-History, echoing our RQ1 findings. 2). $\alpha$ decreases when context is less noisy, as seen in MuSiQue with top-5 related segments compared to top-10. 3). Smaller models rely more on context, with the 0.6B model yielding lower $\alpha$ than the 4B model.

**RQ3: Can in-process structure-aware encoding outperform post-hoc aggregation baselines?**

*1). In-process structure-aware encoding preserves fine-grained contextual information, while post-hoc aggregation might dilutes signals and distorts representations as more segments are added.*

The gap between post-hoc aggregation and individual embedding serves as a good baseline to quantify the value of structural information. When further compared to in-process structure-aware embeddings, in nearly all tasks, interpolating the individual embedding with Struc-Emb variants outperforms post-hoc aggregation. The only exceptions occur when very few related segments are available, as in Cora and Citeseer, where simple pooling does not dilute the signal or introduce noise, and in STaRK-Amazon with Hit@1 and MRR, where pooling preserves the strongest signal most relevant to these metrics. In all other cases, post-hoc aggregation underperforms because averaging over more neighbors dilutes useful information and produces pooled embeddings with numerical values that no longer match the distribution of representations generated by the model. For detail-oriented tasks such as MuSiQue and most metrics in STaRK-Amazon, post-hoc aggregation fails to recover the fine-grained information needed for accurate retrieval.

## 5 CONCLUSION AND FUTURE WORK

We regard this study as preliminary work that demonstrates the potential of incorporating structural information directly into the encoding process, while also acknowledging its limitations. Our experiments focus on the zero-shot setting for initial validation, but fine-tuning or lightweight adaptation could further improve how models process related segments jointly with the target. A key challenge is adapting the model to handle individually encoded parallel caches by analyzing how this shifts attention distributions. Another important direction is to develop training or tuning strategies that improve robustness under noisy context and control balance contextual signals with the target's core semantics. More broadly, by analyzing the benefits and challenges of incorporating structural information into LLM encoders, we hope to provide insights that guide future structure-aware embedding model design and motivate extensions to other modalities such as tabular data (Yan et al.; Wang et al., 2025c; Hegselmann et al., 2023; Fey et al.).

## 6 LLM USAGE DISCLOSURE

We use LLMs solely as writing-assist tools to polish the manuscript. All research ideas, methodology, experiments, analyses, and initial drafts were conceived and written by the authors. LLMs were employed to refine grammar, improve clarity, enhance readability, and in some cases suggest related works for citation.

## 7 REPRODUCIBILITY STATEMENT.

A complete description of our method, baselines, and backbones is provided in Secs. 3.1 and 3.2. Experimental setups, including datasets and evaluation metrics, are reported in Sec. 4.2. Dataset statistics and preprocessing steps are further detailed in Appendix A.1. The complete list of hyperparameters, sensitivity analyses, and additional evaluation results are included in Appendix A.5. Finally, our code is provided in the supplementary material.

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

# A   APPENDIX

## A.1   DATASET DETAILS

**Dataset Processing.**   We provide additional dataset details omitted from the main text, focusing on how target segments are sampled and what text is associated with each segment. The construction of structural relations is described in Sec. 4.2. We also report dataset statistics: retrieval tasks (multi-hop QA and product recommendation) are shown in Table 5, and the remaining datasets are summarized in Table 6.

For clarification of some names of the statistics:

- avg. #tokens target: the average number of tokens in a target segment.
- avg. #tokens related: the average number of tokens in related segments, computed by averaging the lengths of all related segments across all targets.
- avg. sum #tokens related: the average total number of tokens in the full related set for each target.
- #queries w. multi-ans: the number of queries with multiple ground-truth answers in retrieval.

**MuSiQue:** The structural relations are derived from the Wikipedia hyperlink network, constructed from the 2025-04-04 Wikipedia dump using WikiExtractor. The queries originate from the MuSiQue dataset (Trivedi et al., 2022), but since that dataset was based on earlier Wikipedia versions, we filtered out 400 queries to ensure validity under the updated Wikipedia network. Candidate paragraphs come from the original 20 candidates for each filtered query. We retain only those present in the updated Wikipedia data, yielding 7,521 candidate paragraphs. Note that multiple candidates may come from different paragraphs of the same Wikipedia page, as required to support different questions.

For each candidate, we use the original paragraph text from the MuSiQue dataset. Related segments are selected as described in Sec. 4.2. If a related segment is among the candidate paragraphs, we directly use its MuSiQue text. Otherwise, we summarize the corresponding Wikipedia article into a paragraph of approximately 100–150 tokens using GPT-4o-mini to maintain consistent segment length.

**HotpotQA:** We adopt the HotpotQA dataset from MTEB and use the same set of test queries. To construct the candidate pool for retrieval, we first include all ground-truth paragraphs for each question. We then randomly sample additional negative paragraphs from the full set of 5,233,329 paragraphs, using seed 42, to create a total of 30,000 candidate paragraphs. The text associated with each paragraph and query is kept identical to MTEB.

**Citation Networks and E-Commerce Products:** We adopt these 5 datasets from Wang et al. (a) and keep the exact same structural relations and text information.

**STaRK-Amazon:** We adopt the same test set of human and synthetic queries from the original paper (Wu et al., 2024). Candidate products are selected from the full set of product nodes: for each query, we include all ground-truth products and then sample additional negatives. Specifically, we sample $10\times$ the number of queries as negatives for human queries and $5\times$ for synthetic queries, using seed 42. The number of candidates is reported in Table 5. Structural relations follow the original product graph, where neighboring nodes (via get_neighbor_nodes() in the stark_qa library) serve as related segments. Text associated with each node is retrieved using get_doc_info() from the same library.

**Stack Exchange Post Clustering:** This corresponds to the StackExchangeClustering dataset in MTEB. The original dataset contains only the post title and the subsite domain (e.g., math.stackexchange.com) as the clustering label. As described in Sec. 4.2, we additionally use the raw StackExchange dataset[2] , which provides tag information for each post. We construct structural relations by linking two posts if they share all tags, and we filter out posts not present in the raw corpus. Note that sharing a tag does not guarantee the same clustering label (e.g., the tag terminology appears in languagelearning.stackexchange.com and ux.stackexchange.com). For text inputs, we use the same post titles as in MTEB. Following the benchmark protocol, we evaluate on the first two of the original 25 test splits and report the average.

---

[2]`https://huggingface.co/datasets/flax-sentence-embeddings/`
`stackexchange_title_body_jsonl`

Table 5: Dataset statistics for multi-hop retrieval and product recommendation retrieval.

| DATASET | #CANDIDATES | #QUERIES | AVG. #ANSWERS | #QUERIES W. MULTI-ANS | AVG. #RELATED | AVG. #TOKENS TARGET | AVG. #TOKENS RELATED | AVG. SUM #TOKENS RELATED |
|---|---|---|---|---|---|---|---|---|
| MUSIQUE-TOP5-DEGREE | | | | | 5 | | 123.45 | 623.30 |
| MUSIQUE-TOP5-SEMANTIC | | | | | 5 | | 118.26 | 597.41 |
| MUSIQUE-TOP5-PAGERANK | 7521 | 400 | 2.37 | 400 | 5 | 119.03 | 128.71 | 649.86 |
| MUSIQUE-TOP10-DEGREE | | | | | 10 | | 445.87 | 4181.40 |
| MUSIQUE-TOP10-SEMANTIC | | | | | 10 | | 717.77 | 6746.56 |
| MUSIQUE-TOP10-PAGERANK | | | | | 10 | | 420.70 | 3945.37 |
| HOTPOTQA | 30,000 | 7,405 | 2.00 | 7405 | 0.49 | 90.84 | 110.34 | 118.57 |
| STARK-AMAZON-HUMAN | 9,679 | 81 | 19.48 | 64 | 7.31 | 609.89 | 1292.15 | 9778.75 |
| STARK-AMAZON-SYNTH | 91,010 | 1,642 | 5.43 | 1158 | 7.98 | 616.62 | 1347.26 | 11150.37 |

Table 6: Dataset statistics for citation networks, E-commerce networks and StackExchange post clustering datasets.

| DATASET | #SAMPLES | #LABELS | AVG. #RELATED | AVG. #TOKENS TARGET | AVG. #TOKENS RELATED | AVG. SUM #TOKENS RELATED |
|---|---|---|---|---|---|---|
| CORA | 2,708 | 7 | 3.90 | 171.23 | 178.66 | 696.42 |
| CITESEER | 3,186 | 6 | 2.65 | 191.08 | 194.67 | 530.63 |
| PUBMED | 19,717 | 3 | 4.50 | 382.86 | 384.41 | 1728.32 |
| BOOKHIS | 41,551 | 12 | 8.63 | 303.06 | 329.13 | 2924.95 |
| SPORTSFIT | 173,055 | 13 | 10.25 | 31.14 | 32.49 | 345.29 |
| CLUSTERING-SPLIT0 | 10034 | 19 | 4.15 | 13.99 | 14.89 | 197.18 |
| CLUSTERING-SPLIT1 | 10854 | 20 | 1.88 | 13.48 | 14.60 | 123.84 |

**Metric Discussion** For some datasets, we report multiple evaluation metrics, each highlighting different aspects of performance. Below we briefly introduce the metrics used and explain how they emphasize different perspectives.

• *Multi-hop QA Retrieval:* We primarily use nDCG@k, the standard metric in benchmarks such as MTEB. nDCG@k measures how well the ranking places highly relevant items near the top; a perfect ranking yields 1.0, and the score decreases as relevant items are pushed lower. Recall@k is used as a complementary metric: it measures the proportion of relevant items appearing in the top-$k$, focusing on coverage rather than their precise order.

• *Product Recommendation Retrieval:* Following the STaRK dataset, we use Hit@k, Recall@k, and Mean Reciprocal Rank (MRR). Hit@k checks whether at least one relevant item appears in the top-$k$ results, providing a coarse binary view of precision. Recall@k, as above, measures the proportion of relevant items retrieved. MRR computes the reciprocal of the rank of the first relevant item, emphasizing how quickly a relevant result is retrieved. Both MRR and Hit@1 are especially sensitive to representation quality since they only reward models that rank the correct item first.

## A.2 DISCUSSION ON BI-DIRECTIONAL ENCODERS

Our methods align with unidirectional causal encoders, which can reuse cached KV states from related segments as a fixed past context. In contrast, bidirectional encoders like NV-Embed or Embedding-Gemma apply full self-attention across the entire sequence, meaning every token representation depends on both left and right context.

This bidirectional design makes caching context problematic. If related segments are encoded independently, their KV states are incomplete because they are generated without interaction with the target tokens. Using these frozen states later creates an attention mechanism that is inconsistent with the encoder's pretraining, which assumes all tokens are jointly visible. Consequently, caching provides little benefit, as the encoder would need to recompute attention over the combined target and context to remain faithful to its training. Addressing this fundamental mismatch would require significant architectural changes, such as implementing block-sparse attention or retraining the encoder to accept externally cached states.

Even if one imposes sparse attention masks on a bidirectional encoder—placing the target last and restricting flows so that the target attends to all related segments while the related segments do not attend back—the setup still suffers from two limitations. First, most bidirectional encoders rely on pooling strategies such as mean pooling or [CLS] pooling, which mix representations from both target and context tokens. This blurs the separation between target semantics and contextual support, unlike causal encoders where the final target token embedding can be used directly. Second, the masking scheme departs from the bidirectional pretraining distribution, which assumes mutual attention across all tokens. As a result, the attention patterns are distorted, and the model may not aggregate information as intended.

## A.3    CONTEXT INSTRUCTIONS FOR EACH DATASET

**MuSiQue and HotpotQA:** "Summarize the above linked Wikipedia paragraphs of the target paragraph into a contextual representation that captures shared entities, relations, and background knowledge. Use this distilled context, together with the original paragraphs as supporting evidence, when encoding the following target paragraph for retrieval: "

**StackExchange Post Clustering:** "Summarize the above related StackExchange post titles of the target post into a contextual representation that captures overlapping topics, recurring tags, and shared problem domains. Use this distilled context, together with the original posts as supporting evidence, when encoding the following target post for clustering: "

**Cora / Citeseer / Pubmed:** "Summarize the above citing and cited research papers of the target paper into a contextual representation that captures shared domains, recurring methods, and notable overlaps. Use this distilled context, together with the original papers as evidence, when encoding the following target paper for domain classification: "

**Books-History:** "Summarize the above co-purchased or co-viewed history books of the target book into a contextual representation that highlights dominant geographical regions, historical periods, and recurring themes. Use this distilled context, together with the original books as supporting evidence, when encoding the following target book for category classification: "

**Sports-Fitness:** "Summarize the above co-purchased or co-viewed sports & fitness items of the target item into a contextual representation that captures activity types, training goals, and usage contexts. Use this distilled context, together with the original items as supporting evidence, when encoding the following target item for category classification: "

**STaRK-Amazon:** "Summarize the above co-purchased or co-viewed products, brands, colors, and categories of the target product into a contextual representation that captures complementary functions, styles, and usage contexts. Use this distilled context, together with the original attributes as supporting evidence, when encoding the following target product for recommendation: "

## A.4    FURTHER COMPARISON OF STRUC-EMB-SEQ AND STRUC-EMB-PAR

Beyond performance, we compare the three practical aspects that differentiate Struc-Emb-Seq and Struc-Emb-Par: robustness to increasing input length, sensitivity over order of related segments and computation costs.

*1). Performance with increasing text segment length.* Struc-Emb-Seq is constrained by the context window because positional encodings are assigned sequentially, and it also suffers from long-context issues as discussed earlier. To examine how methods scale, we evaluate individual encoding, Struc-Emb-Seq, and Struc-Emb-Par as text segments grow longer. On MuSiQue, we vary segment lengths from about 120 tokens up to 3k, with detailed statistics in Table 7. Specifically, for candidate paragraphs, we expand the text by appending content from the corresponding Wikipedia article until reaching the desired length. For non-candidate related segments, we use GPT-4o-mini to generate summaries of the appropriate length from the full article.

Figure 2 shows that as input length increases, the performance of Struc-Emb-Seq decays faster than Struc-Emb-Par. Around 3k tokens, its accuracy drops below the Struc-Emb-Par variants as the total sequence approaches the 8192-token limit (the setting we adopt following Qwen3 embedding guidelines for efficiency). Individual encoding starts lower but is less affected by text length, since a single document is typically less than 8192 tokens. In contrast, for Struc-Emb-Seq and Struc-Emb-Par, combining related segments with the target makes longer segments results in much longer sequence than the set context window: extended text in each segment reduces the model's ability to integrate other segments, and in some cases the target itself may be truncated when concatenated at the end.

*2). Computation Cost.* We benchmark computation speed for individual encoding, Struc-Emb-Seq, Struc-Emb-Par, and Struc-Emb-Par-Distill using Qwen3-4B on the MuSiQue retrieval task with top-5 related segments (selected by degree) under varying text lengths. For Struc-Emb-Par and Struc-Emb-Par-Distill, we assume related-segment KV caches are precomputed and measure only the time to embed the target while attending to these cached KVs.

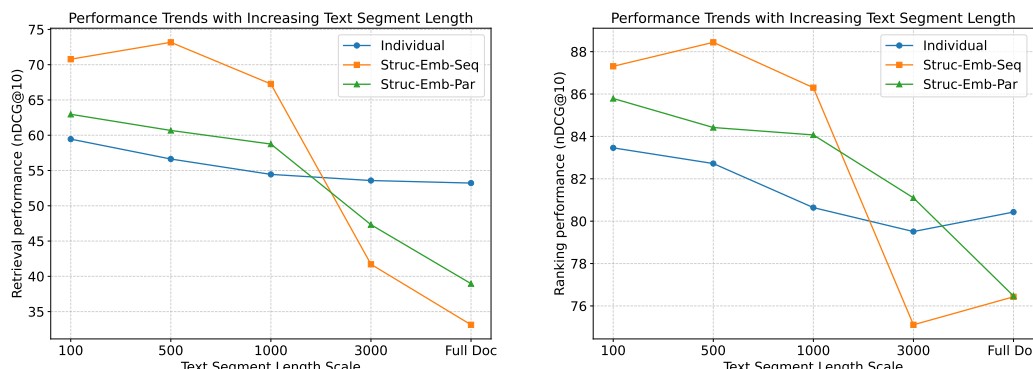

Figure 2: This plot shows the performance trend of Individual encoding, Struc-Emb-Seq and Struc-Emb-Par when we increase the text segment length of both target and related segments.

Table 7: Text length statistics in MuSiQue (top-5 degree setting) under different segment scales. We report three metrics: (i) the average number of tokens in a target segment, (ii) the average number of tokens in related segments, computed by averaging the lengths of all related segments across all targets, and (iii) the average total number of tokens in the related set for each target. Columns 100/500/1000/3000/fulldoc indicate the studied scales, with values showing exact statistics.

| METHOD | 100 | 500 | 1000 | 3000 | FULLDOC |
|---|---|---|---|---|---|
| AVG. #TOKENS TARGET | 119.03 | 503.29 | 806.97 | 1681.90 | 4642.64 |
| AVG. #TOKENS RELATED | 123.45 | 618.35 | 1177.99 | 2935.76 | 10873.93 |
| AVG. SUM #TOKENS RELATED | 623.30 | 3122.02 | 5947.63 | 14772.01 | 54901.97 |

When segment lengths are expanded to 3k (as in Table 7), the the average total tokens per sample exceeds 8,192 tokens—the context limit we initially set. Under this setting, Struc-Emb-Seq is restricted to 8,192 tokens, while Struc-Emb-Par could process up to six times more, leading to an unfair runtime comparison. To ensure parity, we raise the limit to the model's default 32,768 tokens and cap each related/target segment in Struc-Emb-Par methods at 32,768/6 tokens.

As shown in Fig. 3, Struc-Emb-Seq incurs significantly higher computation time than all other methods. Struc-Emb-Par variants are close to individual encoding, with only slight overhead, and Struc-Emb-Par-Distill adds minimal cost over Struc-Emb-Par.

*3). Performance sensitivity to concatenation order.* A key issue in Struc-Emb-Seq is positional bias, which makes performance sensitive to the order of related segments. To test this, we compare three orderings: the default order used when forming relations (results reported in the main text), and two random shuffles (seeds 0 and 42). We evaluate on MuSiQue (top-5 related segments by degree) and on citation networks.

From the Fig. 4, we observe that Struc-Emb-Seq yields significantly different results, whereas Struc-Emb-Par variants remain consistent. When we check the final embedding values from Struc-Emb-Par, there is only difference in embedding with very last few decimal places caused by numerical variance. In MuSiQue, the default order achieves much higher performance than random permutations, since it implicitly reflects relevance: candidate paragraphs from the same question are placed first, followed by higher-scoring related documents, and finally the target. This aligns with previous positional bias study, the model attends to the beginning and the end of the sequence significantly higher than the middle, which aggregates most important information from the closely relevant candidate paragraph and the target segment.

In citation networks, performance varies less across permutations, and the default order carries no strong bias. Still, shuffling related segments produces noticeable deviations, with Citeseer showing the least variance due to its small average number of related segments.

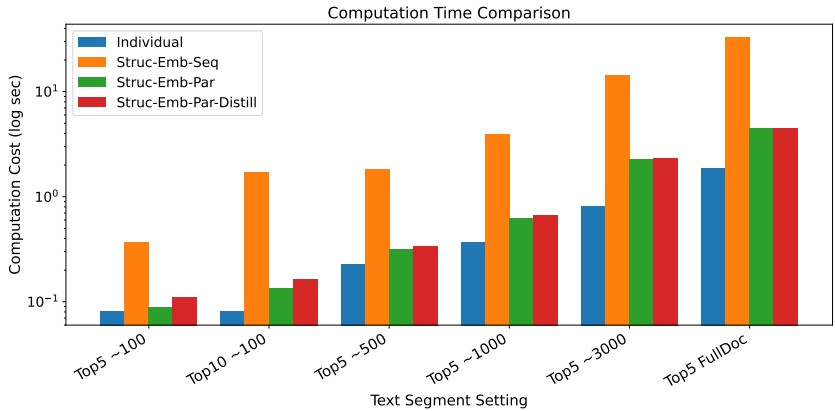

Figure 3: The computation time comparison for different encoding methods under MuSiQue dataset (selection using degree) with varying text length and number of related segments.

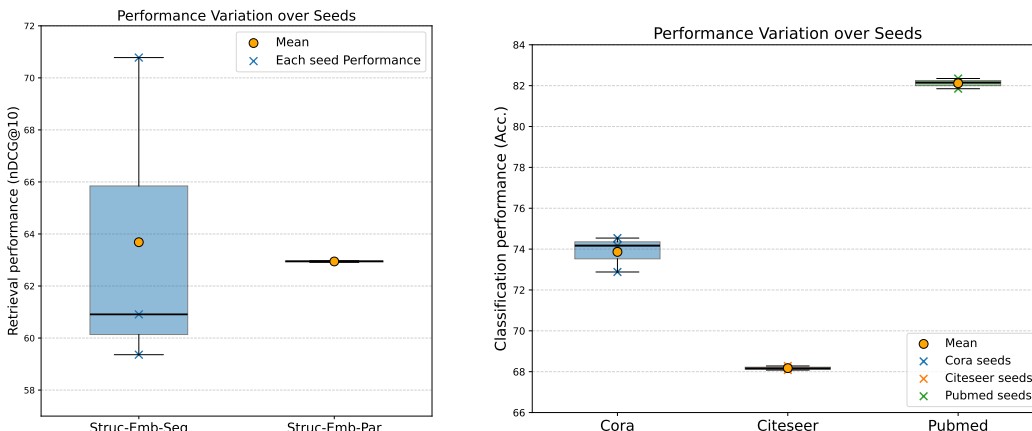

(a) MuSiQue dataset with top5 related segments selected by degree. Comparing Struc-Emb-Seq and Struc-Emb-Par against 3 permutations on the order of related segments.

(b) Struc-Emb-Seq performance variation over 3 permutations on the order of related segments.

Figure 4: Performance variation in Struc-Emb-Seq and Struc-Emb-Par when permuting the order of related segments.

## A.5 ADDITIONAL EXPERIMENTAL RESULTS

### A.5.1 MUSIQUE DATASET EVALUATION WITH RECALL@K

In the main texts for MuSiQue dataset, we include the main metrics nDCG@10 for retrieval and ranking evaluation. Here we include the additional evaluation on Recall@k as metric in Table 8.

### A.5.2 E5-MISTRAL-7B-INSTRUCT RESULTS

Here we will include the additional experimental results on E5-Mistral-7B-Instruct model with MuSiQue dataset in Table 9, Citation and E-commerce classification in Table 10, STaRK-Amazon dataset in Table 11 and Clustering with HotpotQA dataset in Table 12.

In general, the key observations from Qwen3 embedding models hold for the E5-Mistral-7B-Instruct model. Namely, we still observe adding structural information benefit over the text-only individual embeddings. And the benefit from structure-aware embeddings over post-hoc aggregation still holds for almost all datasets and settings. The insight that sequential works with best moderate length noisy context and parallel prefers clean context but can scale well better with long text still applies to E5-Mistral here. Some subtle difference will be discussed later due to the limit in effective context window in E5-Mistral-7B-Instruct. Then, the semantic balancing always help balancing the contextual information against the target semantics. Struc-Emb-Par-Distill still offers some robustness over

Table 8: Results for MuSiQue datasets. "w. balancing" denotes semantic balancing. For each setting (w. and w/balancing), the best method is in **bold** and the second best is underlined. Ret. reports Recall@10 over all candidates, while Rank. reports Recall@5 over candidates per question.

| | METHOD | TOP-5 NEIGHBORS | | | | | | TOP-10 NEIGHBORS | | | | | |
|---|---|---|---|---|---|---|---|---|---|---|---|---|---|
| | | DEGREE | | SEMANTIC | | PAGERANK | | DEGREE | | SEMANTIC | | PAGERANK | |
| | | RET. | RANK. | RET. | RANK. | RET. | RANK. | RET. | RANK. | RET. | RANK. | RET. | RANK. |
| | **QWEN3-EMBEDDING 4B** | | | | | | | | | | | | |
| W/ BALANCING | INDIVIDUAL | 64.60 | 80.83 | 64.60 | 80.83 | 64.60 | 80.83 | 64.60 | 80.83 | 64.60 | 80.83 | 64.60 | 80.83 |
| | STRUC-EMB-SEQ | 81.42 | 88.92 | 83.40 | 88.71 | 81.60 | 88.54 | 70.10 | 83.38 | 71.23 | 84.33 | 70.71 | 83.67 |
| | STRUC-EMB-PAR | 71.48 | 86.35 | 77.21 | 89.00 | 71.35 | 86.98 | 47.60 | 75.10 | 58.33 | 79.54 | 47.65 | 76.10 |
| | STRUC-EMB-PAR-DISTILL | 72.42 | 88.44 | 77.00 | 89.58 | 72.13 | 88.65 | 49.21 | 74.27 | 62.90 | 82.52 | 49.90 | 74.58 |
| W. BALANCING | MEAN NEIGHBOR | 79.04 | 89.94 | 77.73 | 90.06 | 78.56 | 89.94 | 71.00 | 84.88 | 70.88 | 86.83 | 70.52 | 85.42 |
| | WEIGHTED MEAN NEIGHBOR | 80.58 | 90.67 | 76.25 | 89.54 | 80.65 | 90.48 | 71.75 | 85.81 | 69.69 | 86.10 | 71.96 | 85.98 |
| | STRUC-EMB-SEQ | 86.83 | 92.33 | 86.69 | 91.71 | 87.17 | 92.69 | 83.46 | 91.10 | 77.78 | 89.27 | 84.00 | 91.21 |
| | STRUC-EMB-PAR | 80.42 | 89.08 | 80.00 | 90.54 | 79.21 | 89.50 | 70.94 | 84.83 | 68.75 | 84.25 | 70.94 | 85.58 |
| | STRUC-EMB-PAR-DISTILL | 78.98 | 89.31 | 78.38 | 89.88 | 78.69 | 89.77 | 70.15 | 84.23 | 68.83 | 85.54 | 70.29 | 84.90 |
| | **QWEN3-EMBEDDING 0.6B** | | | | | | | | | | | | |
| W/ BALANCING | INDIVIDUAL | 57.40 | 73.75 | 57.40 | 73.75 | 57.40 | 73.75 | 57.40 | 73.75 | 57.40 | 73.75 | 57.40 | 73.75 |
| | STRUC-EMB-SEQ | 73.40 | 85.44 | 75.19 | 85.52 | 73.21 | 85.52 | 60.15 | 79.50 | 59.21 | 78.58 | 59.63 | 79.31 |
| | STRUC-EMB-PAR | 65.13 | 83.42 | 68.19 | 83.56 | 65.10 | 83.81 | 45.38 | 73.31 | 46.63 | 74.58 | 44.63 | 73.40 |
| | STRUC-EMB-PAR-DISTILL | 71.69 | 84.79 | 70.35 | 84.75 | 70.73 | 84.69 | 49.90 | 76.88 | 50.58 | 75.90 | 49.54 | 76.96 |
| W. BALANCING | MEAN NEIGHBOR | 71.88 | 86.73 | 71.04 | 86.71 | 71.52 | 86.33 | 63.29 | 80.06 | 64.73 | 82.25 | 64.98 | 79.81 |
| | WEIGHTED MEAN NEIGHBOR | 75.71 | 88.33 | 68.88 | 85.25 | 74.50 | 88.15 | 65.17 | 81.98 | 63.75 | 81.04 | 57.75 | 81.54 |
| | STRUC-EMB-SEQ | 82.42 | 88.98 | 80.60 | 88.27 | 83.38 | 88.96 | 79.08 | 87.50 | 70.25 | 85.52 | 78.25 | 87.35 |
| | STRUC-EMB-PAR | 74.21 | 85.75 | 72.21 | 85.17 | 73.75 | 85.92 | 66.00 | 80.67 | 61.79 | 79.42 | 65.98 | 80.56 |
| | STRUC-EMB-PAR-DISTILL | 74.44 | 85.71 | 72.92 | 85.02 | 74.31 | 85.92 | 66.02 | 80.33 | 61.90 | 80.13 | 66.06 | 80.65 |
| | **E5-MISTRAL-7B-INSTRUCT** | | | | | | | | | | | | |
| W/ BALANCING | INDIVIDUAL | 61.79 | 75.98 | 61.79 | 75.98 | 61.79 | 75.98 | 61.79 | 75.98 | 61.79 | 75.98 | 61.79 | 75.98 |
| | STRUC-EMB-SEQ | 82.94 | 91.04 | 82.85 | 91.23 | 83.21 | 90.73 | 63.51 | 83.90 | 55.31 | 82.92 | 72.00 | 84.42 |
| | STRUC-EMB-PAR | 65.90 | 86.10 | 72.56 | 88.94 | 65.75 | 85.67 | 48.71 | 76.75 | 60.21 | 82.48 | 49.29 | 76.48 |
| | STRUC-EMB-PAR-DISTILL | 64.19 | 85.65 | 72.65 | 87.79 | 64.44 | 85.94 | 43.79 | 71.15 | 60.04 | 82.94 | 44.38 | 71.63 |
| W. BALANCING | MEAN NEIGHBOR | 78.73 | 89.79 | 76.25 | 89.58 | 77.90 | 89.85 | 72.27 | 85.40 | 72.71 | 87.27 | 72.33 | 85.98 |
| | WEIGHTED MEAN NEIGHBOR | 80.19 | 90.52 | 76.15 | 88.98 | 80.17 | 90.27 | 73.48 | 86.69 | 72.19 | 87.06 | 73.48 | 86.63 |
| | STRUC-EMB-SEQ | 86.96 | 92.13 | 85.85 | 91.94 | 87.04 | 92.50 | 80.81 | 89.65 | 73.08 | 86.65 | 81.08 | 89.96 |
| | STRUC-EMB-PAR | 78.31 | 88.90 | 76.15 | 90.10 | 78.04 | 89.40 | 72.46 | 84.85 | 68.79 | 84.44 | 72.40 | 85.13 |
| | STRUC-EMB-PAR-DISTILL | 78.42 | 89.35 | 76.50 | 89.73 | 77.73 | 89.48 | 72.06 | 84.52 | 69.00 | 84.73 | 72.42 | 84.67 |

Table 9: Results for MuSiQue datasets. "w. balancing" denotes semantic balancing. For each setting (w. and w/balancing), the best method is in **bold** and the second best is underlined. Ret. reports nDCG@10 over all candidates, while Rank. reports nDCG@5 over candidates per question.

| | METHOD | TOP-5 NEIGHBORS | | | | | | TOP-10 NEIGHBORS | | | | | |
|---|---|---|---|---|---|---|---|---|---|---|---|---|---|
| | | DEGREE | | SEMANTIC | | PAGERANK | | DEGREE | | SEMANTIC | | PAGERANK | |
| | | RET. | RANK. | RET. | RANK. | RET. | RANK. | RET. | RANK. | RET. | RANK. | RET. | RANK. |
| | **E5-MISTRAL-7B-INSTRUCT** | | | | | | | | | | | | |
| W/ BALANCING | INDIVIDUAL | 57.14 | 79.83 | 57.14 | 79.83 | 57.14 | 79.83 | 57.14 | 79.83 | 57.14 | 79.83 | 57.14 | 79.83 |
| | STRUC-EMB-SEQ | 73.94 | 88.38 | 75.17 | 89.14 | 74.55 | 88.46 | 63.51 | 83.90 | 55.31 | 82.89 | 64.16 | 84.40 |
| | STRUC-EMB-PAR | 54.67 | 83.27 | 63.21 | 85.70 | 54.62 | 83.31 | 40.45 | 75.97 | 52.03 | 82.09 | 41.38 | 76.30 |
| | STRUC-EMB-PAR-DISTILL | 54.14 | 83.72 | 64.13 | 86.75 | 54.63 | 83.76 | 37.57 | 74.30 | 51.74 | 83.19 | 38.34 | 74.49 |
| W. BALANCING | MEAN NEIGHBOR | 71.10 | 89.22 | 69.03 | 88.03 | 70.84 | 89.12 | 66.02 | 86.37 | 66.30 | 86.34 | 66.05 | 86.56 |
| | WEIGHTED MEAN NEIGHBOR | 72.70 | 89.90 | 68.38 | 87.78 | 72.47 | 89.72 | 66.56 | 86.63 | 65.83 | 86.11 | 66.59 | 86.90 |
| | STRUC-EMB-SEQ | 80.12 | 91.46 | 78.37 | 90.47 | 80.00 | 91.39 | 72.32 | 89.36 | 66.21 | 86.44 | 72.69 | 89.43 |
| | STRUC-EMB-PAR | 70.90 | 88.94 | 69.30 | 87.73 | 70.54 | 88.98 | 66.31 | 86.11 | 63.98 | 85.21 | 66.64 | 86.27 |
| | STRUC-EMB-PAR-DISTILL | 71.30 | 88.97 | 69.50 | 88.40 | 71.26 | 89.10 | 66.18 | 86.27 | 64.29 | 85.40 | 66.86 | 86.45 |

Struc-Emb-Par, but the degree of improvement is less than observed in Qwen3 models, which will be discussed later.

The major distinction from that cause some subtle difference in the results from these two pretrained encoders is their effective context window length. E5-Mistral-7B-Instruct suggests that it is not recommended to have inputs longer than 4096 tokens, so it is naturally designed capture only short context compared to the Qwen3 models. Based on their recommendation, we set the max token to be 4096 in contrast to the 8192 in Qwen3.

Indeed, the performance of E5-Mistral-7B-Instruct is relatively worse on long sequence, like MuSiQue with top10 related segments and STaRK-Amazon datasets with long text input especially under evaluation metrics that rely on more details like Recall@k with larger k. This also explains why some rare cases that the post-hoc aggregation is better since it lacks the capability in capturing long-range dependency from target to each related segments in process with structure-aware encoding. So instead, it is better to encode shorter text individually and do post-hoc aggregation. While in Qwen3, with more inherent capability in capture long dependencies, Struc-Emb variants are better. Regarding moderate length input, their general performance lies between Qwen3 0.6B and 4B.

In addition, we observe that E5-Mistral-7B-Instruct is more robust in attention distribution with respective to the parallel caching. The improvement and degradation from Struc-Emb-Par is more moderate in E5-Mistral-7B compared to Qwen3 models. Also, this tends to diminish the relative improvement from Struc-Emb-Par-Distill on top of Struc-Emb-Par.

Table 10: Results for networked object classification. "w. balancing" denotes semantic balancing. For each setting (w. and w/balancing), the best method is in **bold** and the second best is underlined.

| | METHOD | CORA | | CITESEER | | PUBMED | | BOOKHIS | | SPORTSFIT | |
|---|---|---|---|---|---|---|---|---|---|---|---|
| | | ACC | F1 | ACC | F1 | ACC | F1 | ACC | F1 | ACC | F1 |
| | **E5-MISTRAL-7B-INSTRUCT** | | | | | | | | | | |
| W/ BALANCING | INDIVIDUAL | 71.03 | 66.26 | 65.90 | 61.02 | **83.19** | **82.63** | 60.74 | **26.80** | 66.95 | 53.07 |
| | STRUC-EMB-SEQ | 72.32 | 69.01 | 67.73 | 63.38 | 81.62 | 81.22 | **61.74** | 26.24 | 72.49 | 57.47 |
| | STRUC-EMB-PAR | **73.06** | **70.02** | 68.75 | 64.31 | 81.36 | 80.89 | 60.61 | 26.14 | **72.62** | 57.39 |
| | STRUC-EMB-PAR-DISTILL | 71.59 | 68.03 | **69.06** | **64.63** | 81.34 | 80.74 | 60.38 | 26.10 | 70.78 | 56.32 |
| W. BALANCING | MEAN NEIGHBOR | 74.91 | 71.08 | 69.21 | 64.70 | 83.77 | 83.19 | 61.29 | 26.95 | **73.42** | 60.45 |
| | WEIGHTED MEAN NEIGHBOR | 74.91 | 71.08 | 69.29 | 64.76 | 83.80 | 83.21 | 61.23 | 27.03 | 73.40 | **60.49** |
| | STRUC-EMB-SEQ | **75.28** | 71.20 | 68.98 | 64.71 | **84.05** | **83.55** | **62.22** | **28.28** | 72.67 | 57.93 |
| | STRUC-EMB-PAR | **75.28** | **71.81** | **69.45** | 65.07 | 83.72 | 83.16 | 61.42 | 26.94 | 72.76 | 58.15 |
| | STRUC-EMB-PAR-DISTILL | 74.72 | 70.88 | **69.45** | **65.14** | 83.67 | 83.16 | 61.62 | 27.06 | 71.59 | 57.55 |

Table 11: Results for STaRK-Amazon dataset. "w. balancing" denotes semantic balancing. For each setting (w. and w/balancing), the best method is in **bold** and the second best is underlined.

| | METHOD | HUMAN-GENERATED | | | | | | | SYNTHETIC | | | |
|---|---|---|---|---|---|---|---|---|---|---|---|---|
| | | HIT@1 | HIT@5 | R@20 | R@30 | R@50 | R@90 | MRR | HIT@1 | HIT@5 | R@20 | MRR |
| | **E5-MISTRAL-7B-INSTRUCT** | | | | | | | | | | | |
| W/ BALANCING | INDIVIDUAL | 83.95 | 90.12 | 73.86 | 83.10 | **92.64** | **97.13** | 87.60 | **65.47** | 85.69 | 78.32 | **74.45** |
| | STRUC-EMB-SEQ | 76.54 | 83.95 | 61.76 | 69.15 | 76.53 | 83.63 | 80.34 | 58.77 | 78.38 | 66.89 | 67.82 |
| | STRUC-EMB-PAR | **88.89** | 92.59 | **76.60** | **83.40** | 90.10 | 95.61 | **90.79** | 64.19 | **86.05** | 76.44 | 73.68 |
| | STRUC-EMB-PAR-DISTILL | 86.42 | **93.83** | 74.79 | 82.64 | 89.92 | 96.21 | 90.08 | 61.14 | 83.07 | 74.01 | 70.93 |
| W. BALANCING | MEAN NEIGHBOR | 86.42 | **97.53** | 78.91 | 85.25 | **92.94** | 97.38 | 90.32 | 67.36 | 87.15 | 79.57 | 75.90 |
| | WEIGHTED MEAN NEIGHBOR | 86.42 | **97.53** | 78.95 | 85.25 | 92.91 | 97.43 | 90.15 | 67.42 | 87.09 | 79.50 | 75.94 |
| | STRUC-EMB-SEQ | 90.12 | 96.30 | 78.91 | 85.23 | 92.79 | **97.87** | 92.69 | 67.11 | 87.86 | 79.95 | 76.20 |
| | STRUC-EMB-PAR | 90.12 | 95.06 | 78.62 | 85.27 | 92.73 | 97.33 | 92.14 | **68.21** | 88.37 | **80.28** | **77.01** |
| | STRUC-EMB-PAR-DISTILL | **92.59** | 93.83 | 78.75 | **85.52** | 92.81 | 97.47 | **93.48** | 68.15 | 88.43 | 80.16 | 76.99 |

### A.5.3 $\alpha$ VALUES FOR SEMANTIC BALANCING

As mentioned, we select $\alpha$ based on the samples directly for evaluation using a grid search in range $[0, 1]$ each with difference 0.02. We do this oracle tuning mainly for two reasons: first, it is that not all datasets have some validation datasets for us to perform tuning on this hyperparameter. Second, since we do this oracle searching on evaluation samples for both baseline post-hoc aggregation and structure-aware encoding variants, so we can understand the best and upper limit each method can achieve under semantic balancing, which provides us a better exploration insight that how each dataset task, evaluation metric and encoding method influence the choice of $\alpha$.

We report the optimal $\alpha$ values for all datasets, with MuSiQue dataset in Table 13 for retrieval evaluation and Table 14 for ranking evaluation. The STaRK-Amazon dataset is in Table 16 for human generated queries evaluation and Table 15 for synthetic queries evaluation. The citation network and E-commerce network classification datasets are in Table 17. The stackexchange post clustering and hotpotQA retrieval are in Table 18.

We also calculate the average $\alpha$ values over different encoding methods under the same metric/setting to evaluate how the evaluation and the condition of related segments affect the $\alpha$ selection. Similarly, we calculate the average $\alpha$ values over different metrics/setting under the same encoding method to evaluate how the encoding technique impact the $\alpha$ selection. Based on the results of these $\alpha$ values, we observe the rules as discussed in Sec. 4.3 RQ2.2. In addition, another minor point is that $\alpha$ also reveals encoding design: structure-aware encodings achieve lower $\alpha$ than post-hoc aggregation on the same datasets, showing a more effective encoding of structural information from Struc-Emb variants.

Additionally, we include the plots showing how the performance varies with all $\alpha$ values we search in the range to demonstrate each encoding method's sensitivity to $\alpha$ value for each dataset. We choose the Qwen 0.6B results and select the main metrics in each dataset for demonstration.

In general, the $\alpha$ does not vary a lot in the small range that produce optimal performance. Typically, for each dataset, there is an optimal $\alpha$ range based on its task reliance on the structural information, showing a concave trend or some rather obvious trend of $\alpha$ preference. Then, the optimal $\alpha$ for each encoding method lies in that range, with Struc-Emb encoding have lower value compared to the post-hoc methods. Moreover, most of the time, we can observe a consistent gap in performance within the complete range of $\alpha$ across different encoding method. In other words, the ranking accross encoding methods are rather consistent in nearly all $\alpha$ range we searched over, instead of a highly varies captured trend.

Table 12: Results for StackExchange clustering (Clust.) evaluated with V-measure and HotpotQA dataset. "w. balancing" denotes semantic balancing. For each setting (w. and w/balancing), the best method is in **bold** and the second best is underlined.

| | | | E5-MISTRAL-7B-INSTRUCT | | |
| | METHOD | CLUST. | HOTPOTQA | | |
| | | | NDCG@10 | RECALL@5 | RECALL@10 |
|---|---|---|---|---|---|
| | INDIVIDUAL | 59.43 | 92.29 | 91.74 | 94.02 |
| W/ BALANCING | STRUC-EMB-SEQ | 54.86 | 98.16 | 98.80 | 99.20 |
| | STRUC-EMB-PAR | 54.98 | 98.29 | 98.76 | 99.25 |
| | STRUC-EMB-PAR-DISTILL | **66.17** | **98.46** | **98.92** | **99.34** |
| | MEAN NEIGHBOR | 65.43 | 98.45 | 100 | 100 |
| | WEIGHTED MEAN NEIGHBOR | 65.33 | 98.44 | 100 | 100 |
| W. BALANCING | STRUC-EMB-SEQ | 66.22 | **98.73** | 100 | 100 |
| | STRUC-EMB-PAR | 63.58 | 98.70 | 100 | 100 |
| | STRUC-EMB-PAR-DISTILL | **73.70** | 98.71 | 100 | 100 |

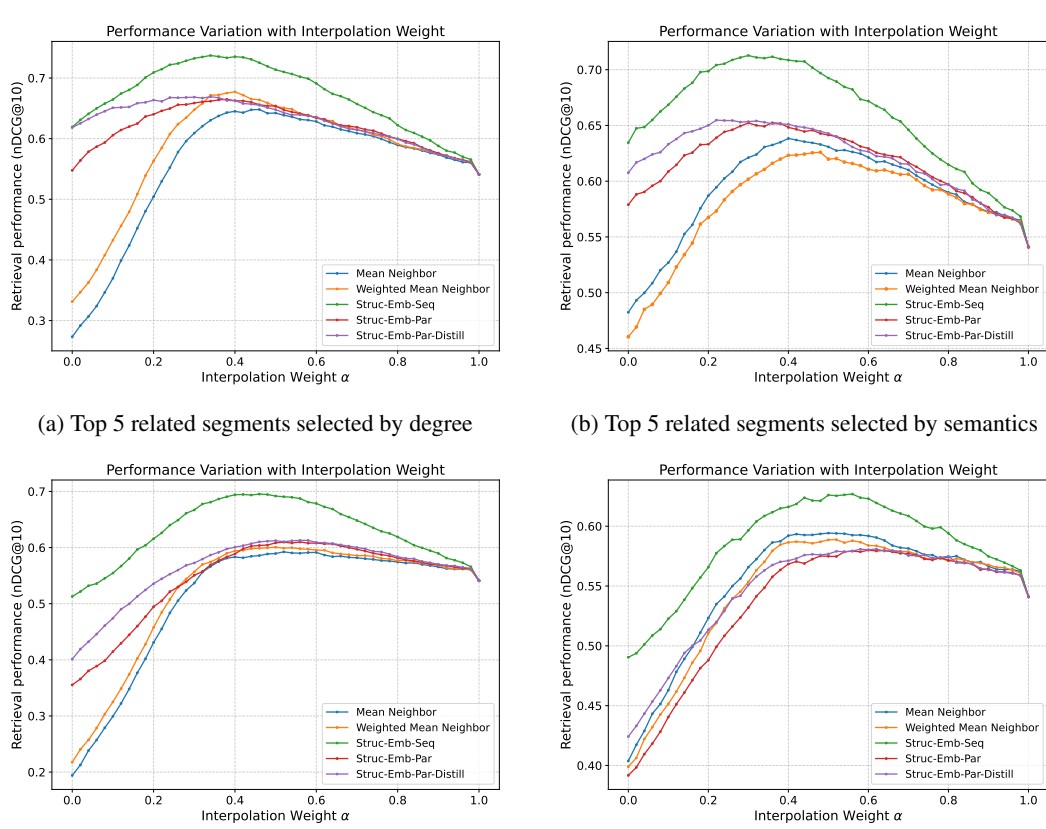

(a) Top 5 related segments selected by degree

(b) Top 5 related segments selected by semantics

(c) Top 10 related segments selected by degree

(d) Top 10 related segments selected by semantics

Figure 5: $\alpha$ sensitivity study for MuSiQue datasets

In addition, the performance curve with varying $\alpha$ is smooth for most of the datasets, except for some datasets with very few related segments that does not show very strong reliance on structural information, like cora, citeseer and clustering. For different metrics, the trend is mostly aligned, with some small difference in the smoothness of the curve, like some Acc is more smooth than F1 since it is less sensitive. Similarly, MRR score that depends only on the rank of the first relevant item is more sensitive compared to the Recall that measured on a wider range.

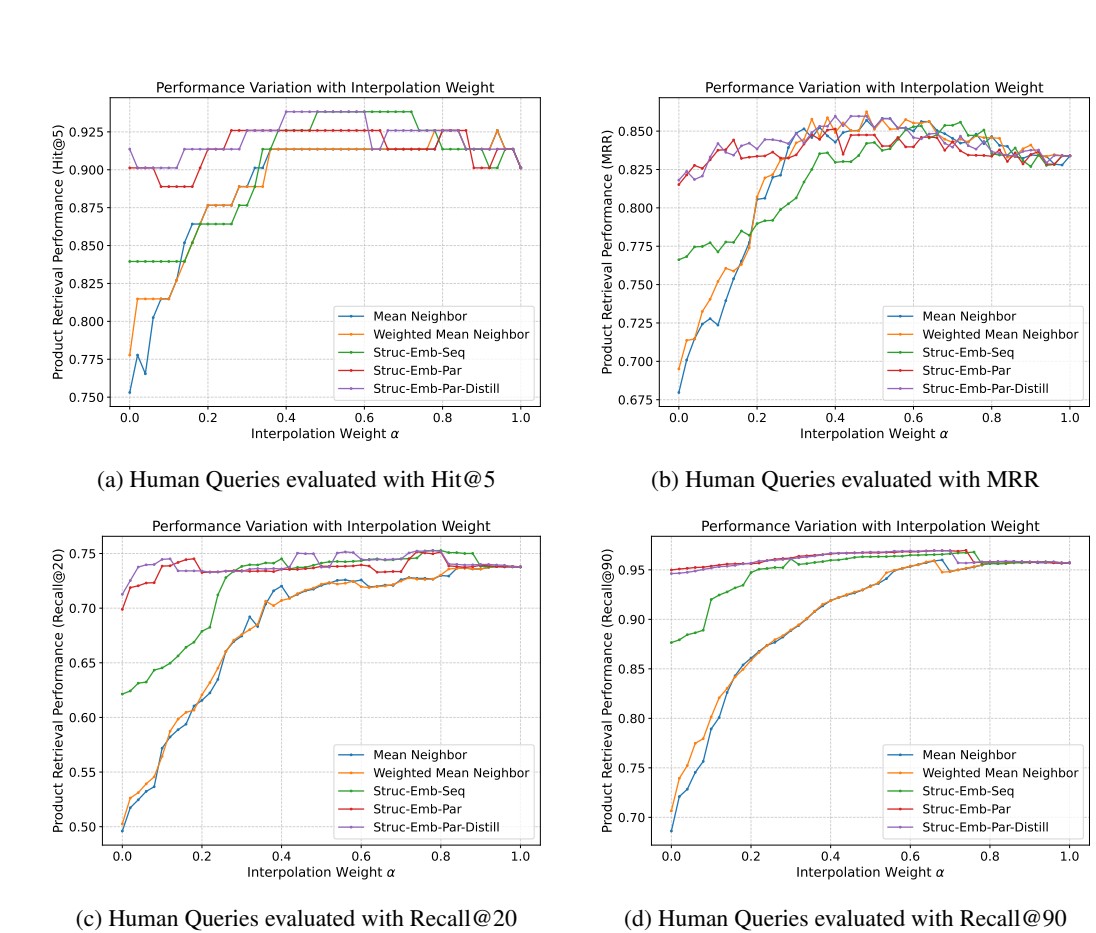

(a) Human Queries evaluated with Hit@5

(b) Human Queries evaluated with MRR

(c) Human Queries evaluated with Recall@20

(d) Human Queries evaluated with Recall@90

Figure 6: $\alpha$ sensitivity study for STaRK-Amazon with human-generated queries

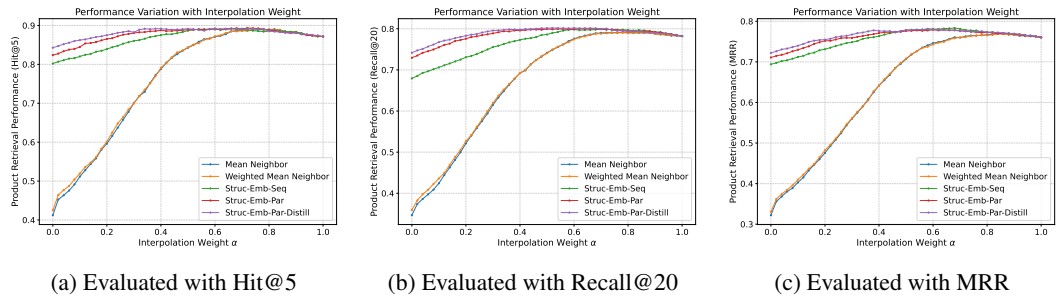

(a) Evaluated with Hit@5

(b) Evaluated with Recall@20

(c) Evaluated with MRR

Figure 7: $\alpha$ sensitivity study for STaRK-Amazon datasets with synthetic queries

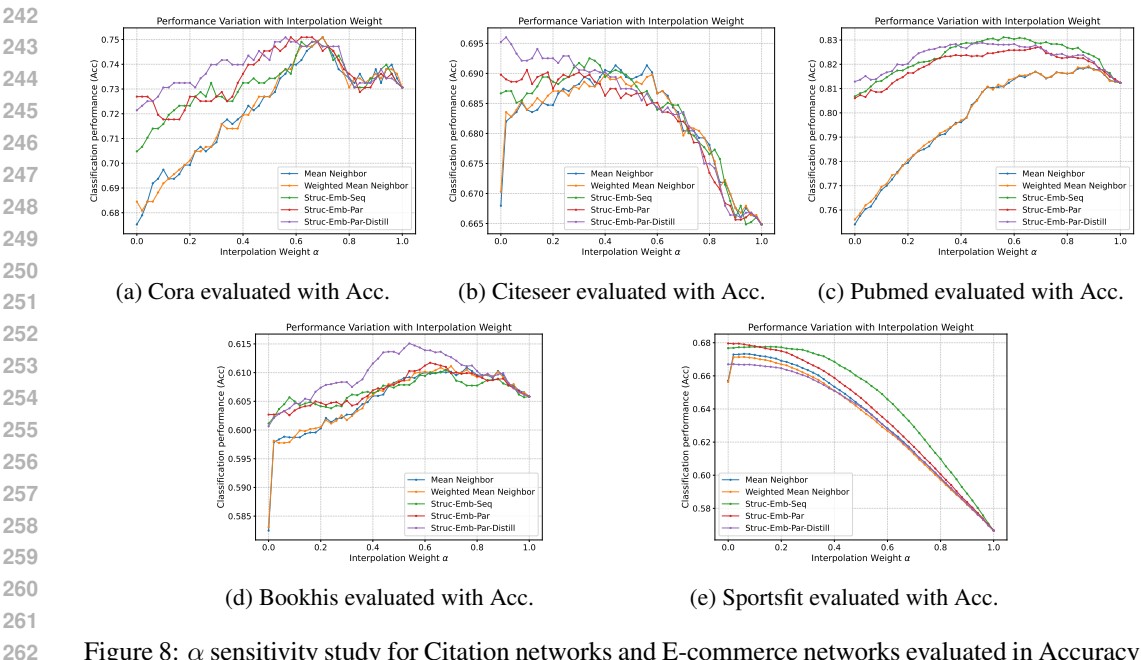

(a) Cora evaluated with Acc.     (b) Citeseer evaluated with Acc.     (c) Pubmed evaluated with Acc.

(d) Bookhis evaluated with Acc.     (e) Sportsfit evaluated with Acc.

Figure 8: $\alpha$ sensitivity study for Citation networks and E-commerce networks evaluated in Accuracy

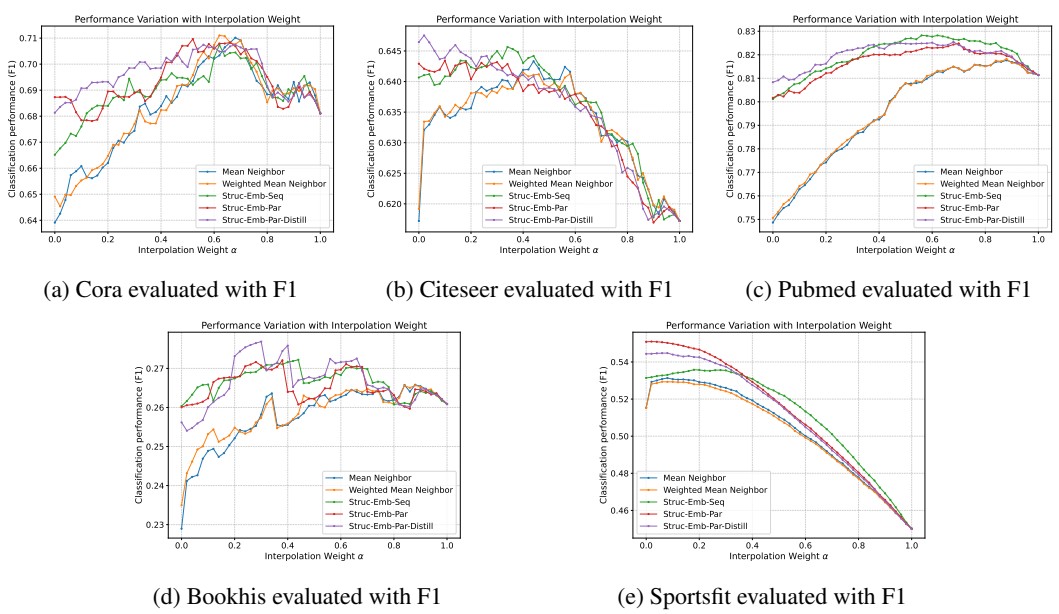

(a) Cora evaluated with F1     (b) Citeseer evaluated with F1     (c) Pubmed evaluated with F1

(d) Bookhis evaluated with F1     (e) Sportsfit evaluated with F1

Figure 9: $\alpha$ sensitivity study for Citation networks and E-commerce networks evaluated in F1 score

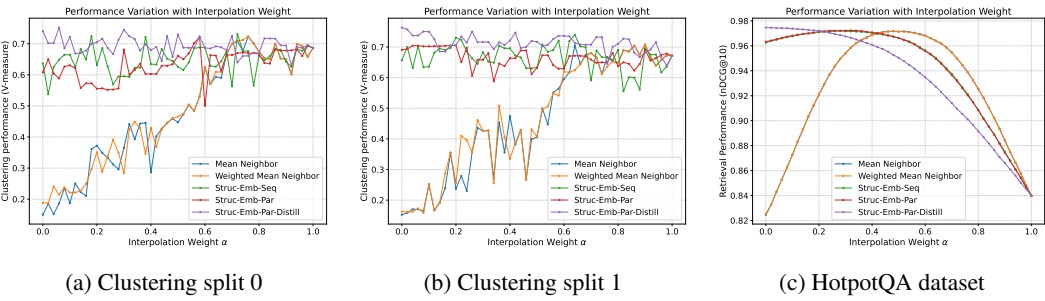

(a) Clustering split 0     (b) Clustering split 1     (c) HotpotQA dataset

Figure 10: $\alpha$ sensitivity study for Clustering datasets and HotpotQA datasets

Table 13: This table reports the optimal $\alpha$ to interpolate the target and structure-aware embedding over MuSiQue datasets evaluated under retrieval using nDCG@10 as metrics. The Avg. over cases refers to the average $\alpha$ over different selection criterion of related segments for the same method and Avg. over methods refers to the average $\alpha$ over different methods under the same selection criterion.

| Method | Top-5 Neighbors | | | Top-10 Neighbors | | | Avg. over cases |
|---|---|---|---|---|---|---|---|
| | Degree | Semantic | Pagerank | Degree | Semantic | Pagerank | |
| **Qwen 0.6B Retrieval nDCG@10** | | | | | | | |
| mean neighbor | 0.46 | 0.40 | 0.44 | 0.52 | 0.50 | 0.52 | 0.473 |
| weighted mean neighbor | 0.40 | 0.48 | 0.40 | 0.50 | 0.52 | 0.54 | 0.473 |
| Struc-Emb-Seq | 0.34 | 0.30 | 0.34 | 0.46 | 0.56 | 0.46 | 0.410 |
| Struc-Emb-Par | 0.38 | 0.36 | 0.38 | 0.56 | 0.60 | 0.58 | 0.477 |
| Struc-Emb-Par-Distill | 0.34 | 0.22 | 0.32 | 0.56 | 0.62 | 0.54 | 0.433 |
| Avg. over methods | 0.384 | 0.352 | 0.376 | 0.520 | 0.560 | 0.528 | 0.453 |
| **Qwen 4B Retrieval nDCG@10** | | | | | | | |
| mean neighbor | 0.42 | 0.42 | 0.42 | 0.46 | 0.54 | 0.50 | 0.460 |
| weighted mean neighbor | 0.42 | 0.42 | 0.42 | 0.48 | 0.60 | 0.46 | 0.467 |
| Struc-Emb-Seq | 0.32 | 0.28 | 0.40 | 0.44 | 0.42 | 0.44 | 0.383 |
| Struc-Emb-Par | 0.36 | 0.26 | 0.36 | 0.60 | 0.66 | 0.60 | 0.473 |
| Struc-Emb-Par-Distill | 0.38 | 0.24 | 0.38 | 0.60 | 0.60 | 0.54 | 0.457 |
| Avg. over methods | 0.380 | 0.324 | 0.396 | 0.516 | 0.564 | 0.508 | 0.448 |
| **E5-Mistral-7B Retrieval nDCG@10** | | | | | | | |
| mean neighbor | 0.46 | 0.46 | 0.46 | 0.54 | 0.46 | 0.54 | 0.487 |
| weighted mean neighbor | 0.42 | 0.44 | 0.44 | 0.48 | 0.46 | 0.54 | 0.463 |
| Struc-Emb-Seq | 0.30 | 0.34 | 0.30 | 0.42 | 0.50 | 0.34 | 0.367 |
| Struc-Emb-Par | 0.42 | 0.32 | 0.38 | 0.46 | 0.48 | 0.54 | 0.433 |
| Struc-Emb-Par-Distill | 0.44 | 0.34 | 0.36 | 0.52 | 0.54 | 0.52 | 0.453 |
| Avg. over methods | 0.408 | 0.380 | 0.388 | 0.484 | 0.488 | 0.496 | 0.441 |

Table 14: This table reports the optimal $\alpha$ to interpolate the target and structure-aware embedding over MuSiQue datasets evaluated under ranking using nDCG@10 as metrics. The Avg. over cases refers to the average $\alpha$ over different selection criterion of related segments for the same method and Avg. over methods refers to the average $\alpha$ over different methods under the same selection criterion.

| Method | Top-5 Neighbors | | | Top-10 Neighbors | | | Avg. over cases |
|---|---|---|---|---|---|---|---|
| | Degree | Semantic | Pagerank | Degree | Semantic | Pagerank | |
| **Qwen 0.6B Ranking nDCG@10** | | | | | | | |
| mean neighbor | 0.38 | 0.34 | 0.40 | 0.46 | 0.36 | 0.44 | 0.397 |
| weighted mean neighbor | 0.40 | 0.36 | 0.40 | 0.44 | 0.42 | 0.46 | 0.413 |
| Struc-Emb-Seq | 0.40 | 0.44 | 0.38 | 0.50 | 0.44 | 0.52 | 0.447 |
| Struc-Emb-Par | 0.42 | 0.36 | 0.38 | 0.46 | 0.40 | 0.44 | 0.410 |
| Struc-Emb-Par-Distill | 0.28 | 0.26 | 0.22 | 0.44 | 0.42 | 0.46 | 0.347 |
| Avg. over methods | 0.376 | 0.352 | 0.356 | 0.460 | 0.408 | 0.464 | 0.403 |
| **Qwen 4B Ranking nDCG@10** | | | | | | | |
| mean neighbor | 0.40 | 0.34 | 0.42 | 0.46 | 0.40 | 0.44 | 0.410 |
| weighted mean neighbor | 0.40 | 0.40 | 0.42 | 0.44 | 0.54 | 0.42 | 0.437 |
| Struc-Emb-Seq | 0.40 | 0.38 | 0.42 | 0.44 | 0.46 | 0.48 | 0.430 |
| Struc-Emb-Par | 0.32 | 0.30 | 0.32 | 0.42 | 0.56 | 0.40 | 0.387 |
| Struc-Emb-Par-Distill | 0.28 | 0.18 | 0.30 | 0.54 | 0.50 | 0.50 | 0.383 |
| Avg. over methods | 0.360 | 0.320 | 0.376 | 0.460 | 0.492 | 0.448 | 0.409 |
| **E5-Mistral-7B Ranking nDCG@10** | | | | | | | |
| mean neighbor | 0.42 | 0.40 | 0.42 | 0.44 | 0.38 | 0.44 | 0.417 |
| weighted mean neighbor | 0.40 | 0.34 | 0.40 | 0.48 | 0.42 | 0.48 | 0.420 |
| Struc-Emb-Seq | 0.26 | 0.28 | 0.36 | 0.42 | 0.44 | 0.42 | 0.363 |
| Struc-Emb-Par | 0.32 | 0.32 | 0.34 | 0.50 | 0.44 | 0.44 | 0.393 |
| Struc-Emb-Par-Distill | 0.34 | 0.28 | 0.36 | 0.52 | 0.40 | 0.50 | 0.400 |
| Avg. over methods | 0.348 | 0.324 | 0.376 | 0.472 | 0.416 | 0.456 | 0.399 |

Table 15: This table reports the optimal $\alpha$ to interpolate the target and structure-aware embedding over STaRK-Amazon datasets with synthetic queries. The Avg. refers to the average $\alpha$ over different metrics for the same method and Avg. over methods refers to the average $\alpha$ over different methods under the same evaluation metric.

| Method | Qwen 0.6B | | | | | Qwen 4B | | | | | E5-Mistral-7B | | | | |
|---|---|---|---|---|---|---|---|---|---|---|---|---|---|---|---|
| | Hit@1 | Hit@5 | Recall@20 | MRR | Avg | Hit@1 | Hit@5 | Recall@20 | MRR | Avg | Hit@1 | Hit@5 | Recall@20 | MRR | Avg |
| mean neighbor | 0.50 | 0.54 | 0.60 | 0.52 | 0.54 | 0.92 | 0.78 | 0.86 | 0.92 | 0.87 | 0.74 | 0.70 | 0.68 | 0.74 | 0.715 |
| weighted mean neighbor | 0.84 | 0.76 | 0.74 | 0.84 | 0.795 | 0.92 | 0.78 | 0.86 | 0.82 | 0.845 | 0.74 | 0.78 | 0.66 | 0.74 | 0.730 |
| Struc-Emb-Seq | 0.68 | 0.60 | 0.70 | 0.68 | 0.665 | 0.70 | 0.64 | 0.80 | 0.70 | 0.71 | 0.80 | 0.80 | 0.74 | 0.80 | 0.785 |
| Struc-Emb-Par | 0.64 | 0.72 | 0.62 | 0.60 | 0.645 | 0.52 | 0.42 | 0.56 | 0.62 | 0.53 | 0.60 | 0.66 | 0.60 | 0.60 | 0.615 |
| Struc-Emb-Par-Distill | 0.58 | 0.34 | 0.52 | 0.58 | 0.505 | 0.52 | 0.64 | 0.64 | 0.52 | 0.58 | 0.64 | 0.60 | 0.60 | 0.64 | 0.620 |
| Avg. over methods | 0.648 | 0.592 | 0.636 | 0.644 | 0.630 | 0.716 | 0.652 | 0.744 | 0.716 | 0.707 | 0.704 | 0.708 | 0.656 | 0.704 | 0.693 |

Table 16: This table reports the optimal $\alpha$ to interpolate the target and structure-aware embedding over STaRK-Amazon datasets with human-generated queries. The Avg. over metrics refers to the average $\alpha$ over different metrics for the same method and Avg. over methods refers to the average $\alpha$ over different methods under the same evaluation metric.

| Method | Hit@1 | Hit@5 | Recall@20 | Recall@30 | Recall@50 | Recall@90 | MRR | Avg over metrics |
|---|---|---|---|---|---|---|---|---|
| **Qwen 0.6B** | | | | | | | | |
| mean neighbor | 0.30 | 0.80 | 0.98 | 0.46 | 0.82 | 0.68 | 0.52 | 0.651 |
| weighted mean neighbor | 0.34 | 0.78 | 0.94 | 0.58 | 0.84 | 0.66 | 0.48 | 0.660 |
| Struc-Emb-Seq | 0.72 | 0.48 | 0.80 | 0.40 | 0.64 | 0.76 | 0.72 | 0.646 |
| Struc-Emb-Par | 0.14 | 0.26 | 0.74 | 0.28 | 0.60 | 0.74 | 0.40 | 0.451 |
| Struc-Emb-Par-Distill | 0.40 | 0.40 | 0.78 | 0.70 | 0.48 | 0.66 | 0.44 | 0.551 |
| Avg. over methods | 0.380 | 0.544 | 0.848 | 0.484 | 0.676 | 0.700 | 0.512 | 0.592 |
| **Qwen 4B** | | | | | | | | |
| mean neighbor | 0.94 | 0.62 | 0.92 | 0.90 | 0.76 | 0.92 | 0.94 | 0.857 |
| weighted mean neighbor | 0.72 | 0.60 | 0.92 | 0.90 | 0.74 | 0.92 | 0.74 | 0.791 |
| Struc-Emb-Seq | 0.68 | 0.38 | 0.86 | 0.56 | 0.40 | 0.90 | 0.68 | 0.637 |
| Struc-Emb-Par | 0.30 | 0.00 | 0.76 | 0.84 | 0.30 | 0.84 | 0.32 | 0.480 |
| Struc-Emb-Par-Distill | 0.54 | 0.08 | 0.92 | 0.78 | 0.48 | 0.76 | 0.54 | 0.586 |
| Avg. over methods | 0.636 | 0.336 | 0.876 | 0.796 | 0.536 | 0.868 | 0.644 | 0.670 |
| **E5-Mistral-7B** | | | | | | | | |
| mean neighbor | 0.74 | 0.50 | 0.66 | 0.72 | 0.84 | 0.86 | 0.48 | 0.686 |
| weighted mean neighbor | 0.74 | 0.46 | 0.66 | 0.72 | 0.86 | 0.86 | 0.74 | 0.720 |
| Struc-Emb-Seq | 0.52 | 0.46 | 0.88 | 0.88 | 0.94 | 0.74 | 0.52 | 0.706 |
| Struc-Emb-Par | 0.20 | 0.14 | 0.84 | 0.76 | 0.96 | 0.94 | 0.20 | 0.577 |
| Struc-Emb-Par-Distill | 0.14 | 0.00 | 0.76 | 0.74 | 0.82 | 0.74 | 0.22 | 0.489 |
| Avg. over methods | 0.468 | 0.312 | 0.760 | 0.764 | 0.884 | 0.828 | 0.432 | 0.635 |

Table 17: Node classification performance (Accuracy and F1) across citation (Cora, Citeseer, Pubmed) and text classification (Bookhis, Sportsfit) datasets. Reported results compare neighbor aggregation baselines and proposed methods.

| | Acc | | | | | F1 | | | | |
|---|---|---|---|---|---|---|---|---|---|---|
| Method | Cora | Citeseer | Pubmed | Bookhis | Sportsfit | Cora | Citeseer | Pubmed | Bookhis | Sportsfit |
| **Qwen 0.6B** | | | | | | | | | | |
| mean neighbor | 0.70 | 0.44 | 0.88 | 0.76 | 0.06 | 0.68 | 0.44 | 0.88 | 0.88 | 0.08 |
| weighted mean neighbor | 0.70 | 0.40 | 0.88 | 0.70 | 0.06 | 0.62 | 0.40 | 0.88 | 0.88 | 0.06 |
| Struc-Emb-Seq | 0.62 | 0.34 | 0.56 | 0.68 | 0.14 | 0.62 | 0.34 | 0.56 | 0.44 | 0.18 |
| Struc-Emb-Par | 0.58 | 0.10 | 0.70 | 0.62 | 0.00 | 0.52 | 0.10 | 0.70 | 0.34 | 0.02 |
| Struc-Emb-Par-Distill | 0.56 | 0.02 | 0.48 | 0.54 | 0.02 | 0.56 | 0.02 | 0.68 | 0.30 | 0.08 |
| Avg. over methods | 0.632 | 0.26 | 0.70 | 0.66 | 0.056 | 0.600 | 0.26 | 0.740 | 0.568 | 0.084 |
| **Qwen 4B** | | | | | | | | | | |
| mean neighbor | 0.74 | 0.48 | 0.86 | 0.72 | 0.02 | 0.74 | 0.48 | 0.86 | 0.78 | 0.10 |
| weighted mean neighbor | 0.64 | 0.46 | 0.86 | 0.70 | 0.02 | 0.64 | 0.46 | 0.86 | 0.42 | 0.06 |
| Struc-Emb-Seq | 0.58 | 0.64 | 0.70 | 0.68 | 0.00 | 0.60 | 0.64 | 0.70 | 0.56 | 0.28 |
| Struc-Emb-Par | 0.80 | 0.72 | 0.70 | 0.66 | 0.00 | 0.44 | 0.72 | 0.70 | 0.44 | 0.00 |
| Struc-Emb-Par-Distill | 0.74 | 0.34 | 0.78 | 0.64 | 0.00 | 0.74 | 0.34 | 0.82 | 0.40 | 0.06 |
| Avg. over methods | 0.700 | 0.528 | 0.780 | 0.680 | 0.008 | 0.632 | 0.528 | 0.788 | 0.520 | 0.100 |
| **E5-Mistral-7B** | | | | | | | | | | |
| mean neighbor | 0.52 | 0.30 | 0.84 | 0.68 | 0.04 | 0.52 | 0.42 | 0.84 | 0.90 | 0.08 |
| weighted mean neighbor | 0.52 | 0.42 | 0.84 | 0.68 | 0.02 | 0.52 | 0.42 | 0.84 | 0.40 | 0.08 |
| Struc-Emb-Seq | 0.50 | 0.30 | 0.60 | 0.34 | 0.14 | 0.34 | 0.30 | 0.60 | 0.32 | 0.36 |
| Struc-Emb-Par | 0.24 | 0.22 | 0.72 | 0.58 | 0.18 | 0.24 | 0.22 | 0.72 | 0.90 | 0.2 |
| Struc-Emb-Par-Distill | 0.38 | 0.36 | 0.74 | 0.60 | 0.26 | 0.40 | 0.36 | 0.74 | 0.92 | 0.38 |
| Avg. over methods | 0.432 | 0.32 | 0.748 | 0.576 | 0.128 | 0.404 | 0.344 | 0.748 | 0.688 | 0.220 |

Table 18: Retrieval and clustering evaluation on HotpotQA and clustering splits. We report performance for different neighbor aggregation methods and structure-aware variants across Qwen 0.6B, Qwen 4B, and E5-Mistral-7B.

| | Qwen 0.6B | | | | Qwen 4B | | | | E5-Mistral-7B | | | |
|---|---|---|---|---|---|---|---|---|---|---|---|---|
| Method | HotpotQA | Clust. split0 | Clust. split1 | Clus. Avg | HotpotQA | Clust. split0 | Clust. split1 | Clus. Avg | HotpotQA | Clust. split0 | Clust. split1 | Clus. Avg |
| mean neighbor | 0.48 | 0.72 | 0.74 | 0.73 | 0.46 | 0.76 | 0.90 | 0.83 | 0.52 | 0.78 | 0.96 | 0.87 |
| weighted mean neighbor | 0.48 | 0.72 | 0.74 | 0.73 | 0.46 | 0.76 | 0.90 | 0.83 | 0.52 | 0.78 | 0.96 | 0.87 |
| Struc-Emb-Seq | 0.28 | 0.42 | 0.34 | 0.38 | 0.38 | 0.72 | 0.64 | 0.68 | 0.30 | 0.62 | 0.14 | 0.38 |
| Struc-Emb-Par | 0.32 | 0.72 | 0.34 | 0.53 | 0.34 | 0.58 | 0.20 | 0.39 | 0.24 | 0.72 | 0.76 | 0.74 |
| Struc-Emb-Par-Distill | 0.00 | 0.52 | 0.88 | 0.70 | 0.20 | 0.06 | 0.00 | 0.03 | 0.26 | 0.62 | 0.60 | 0.61 |
| Avg. over methods | 0.312 | 0.620 | 0.608 | 0.614 | 0.368 | 0.576 | 0.528 | 0.552 | 0.368 | 0.704 | 0.684 | 0.694 |

