# OpenReview forum: "Struc-EMB: The Potential of Structure-Aware Encoding in Language Embeddings"
_ICLR.cc/2026/Conference — Submitted to ICLR 2026_

### Official Review · Reviewer_CJha · 2025-10-29

**Soundness:** 3
**Presentation:** 2
**Contribution:** 2
**Rating:** 6
**Confidence:** 4

**Summary:**

The paper studies “structure-aware” text embeddings by injecting graph/relational context (e.g., linked neighbors, co-occurrence, citations) during encoding rather than aggregating representations post-hoc. Two inference-time, training-free variants are proposed: a sequential concatenation strategy (Seq) that encodes the target text with retrieved neighbors as a single sequence, and a parallel key–value strategy (Par) that supplies neighbor representations as external attention sources without serial concatenation. Extensive zero-shot evaluations across retrieval, clustering, classification, and recommendation show that structure-aware encoding generally outperforms both plain “individual” encodings and after-the-fact pooling.

**Strengths:**

- Clear articulation of “encoding-time structure injection” vs post-hoc aggregation, with two complementary designs (Seq vs Par) and principled discussion of trade-offs (robustness to noise, long-context behavior, and compute).
- Evaluations span multiple tasks/datasets and report consistent gains for structure-aware embeddings relative to individual encodings and standard aggregation baselines.
- Methods are training-free at inference time and thus easy to bolt onto existing embedders; simple techniques (context distillation, semantic balancing) are effective in practice.

**Weaknesses:**

- Positioning vs encode-time RAG/PRF is under-specified (mechanism-level overlap).
1. The Seq pathway is closely related to retrieval-augmented fusion that injects evidence during encoding/decoding, e.g., Fusion-in-Decoder (FiD) [1], REALM [2], RETRO [3], and black-box prepend methods such as REPLUG [4]. PRF-style contextual expansion with BERT encoders (BERT-QE) [5] also fuses feedback text at encoding time.
2. The Par pathway resembles dense-retrieval PRF that augments the query with additional feedback vectors/keys in parallel, e.g., ANCE-PRF [6], subsequent reproductions/enhancements [7], multi-representation PRF [8], and ColBERT-PRF/FairPRF [9].

Although this paper targets general-purpose embeddings rather than generation/reranking, the encode-time fusion mechanisms are substantially similar. A more systematic, controlled comparison is needed (same backbone, retrieval pool, neighbor budget, and fusion budget) to sharpen novelty claims.

- If any interpolation/weighting hyperparameters are tuned on evaluation splits, that inflates reported effectiveness vs deployment-realistic settings; stricter selection (validation only) or self-tuning heuristics would strengthen claims.

[1] Izacard & Grave. Leveraging Passage Retrieval with Generative Models (Fusion-in-Decoder). 2021.

[2] Guu et al. REALM: Retrieval-Augmented Language Model Pre-Training. ICML 2020.

[3] Borgeaud et al. RETRO: Improving Language Models by Retrieving from Trillions of Tokens. PMLR 2022.

[4] Shi et al. REPLUG: Retrieval-Augmented Black-Box Language Models. NAACL 2024.

[5] Zheng et al. BERT-QE: Contextualized Query Expansion for Document Re-Ranking. Findings of EMNLP 2020.

[6] Yu, Xiong, Callan. ANCE-PRF: Improving Query Representations for Dense Retrieval with Pseudo-Relevance Feedback. CIKM 2021.

[7] Li et al. Improving Query Representations for Dense Retrieval with PRF: A Reproducibility Study. ECIR 2022.

[8] Wang et al. Pseudo-Relevance Feedback for Multiple Representation Dense Retrieval. arXiv 2021.

[9] Jaenich et al. ColBERT-FairPRF: Towards Fair PRF in Multiple-Representation Dense Retrieval. ECIR 2023.

**Questions:**

None

---

> ### Author Response · Authors · 2025-11-22
>
> We thank the reviewer for recognizing the motivation and the design of our proposed methods and the comprehensive evaluation. Below are the responses to the reviewers comments.
>
> > 1). Positioning vs. encode-time RAG/PRF is under-specified (Weakness 1)
>
> We thank the reviewer for providing this list of highly relevant work. We agree there is a mechanical overlap (e.g., concatenation, attention) with these methods. However, we respectfully argue that our work's novelty is established by a **fundamentally different task, context, and set of technical challenges**. And we will add the discussion regarding mentioned works to our paper as further related work discussion.
>
> - **Task: Query-Independent Embedding vs. Query-Specific Output:** The RAG/PRF methods cited (e.g., FiD, REALM, ColBERT-PRF) are designed for tasks like generation or re-ranking. Their goal is to fuse information to produce a query-specific output (an answer or a score). In contrast, our goal is to produce a **general-purpose, query-independent but structure-augmented text embedding**. Therefore, it is worth a systematic investigation under the new setting since latest LLM encoders have different objectives and inductive biases which might lead to different insights.
> - **Context: Predefined Structure vs. Retrieved Relevance:** RAG/PRF operates on retrieved documents, which are assumed to be semantically relevant to a query. Our work operates on **predefined structural neighbors** (e.g., citations, co-purchase links), which may be **structurally related but semantically noisy** or "distracting." This distinct problem space is what motivated our novel contributions—**Context Distillation and Semantic Balancing**—which are specifically designed to manage this type of noise when generating embedding.
> - **Mechanism: Non-Trivial Adaptations for Embedding:** While **Struc-Emb-Seq** (concatenation) is a straightforward method, **Struc-Emb-Par** (parallel caching) required **non-trivial technical design**. It is not a simple reuse of existing parallel caching mechanisms, but tailored to generate structure-aware embedding that effectively gathers information from multiple related segments simultaneously. Specifically, we had to:
>     - **Re-allocate Positional Encodings (PE):** Assign all neighbor KVs to a shared PE range to mitigate positional bias.
>     - **Modify Attention:** Adapt the attention mechanism to ground sparse attention in the predefined structural relations, enabling a "message-passing-like" aggregation of KV caches.
>
> Finally, a primary contribution of our paper is the **first systematic study of these two classes of in-process mechanisms for embedding generation**. We are not just proposing an algorithm; we are establishing the landscape and analyzing the trade-offs between these two approaches, which we believe is a significant contribution.
>
> > 2). Interpolation hyperparameter tuned on evaluation split (Weakness 2)
>
> We thank the reviewer for this sharp and critical methodological question. We want to be fully transparent: the interpolation hyperparameter ($\alpha$) was indeed tuned on the evaluation set in the paper. This was done **uniformly for all methods**, including all post-hoc baselines to conduct a fair study comparing the **upper-bound potential** of each method under optimal conditions.
>
> In the paper, we also **validated the robustness of $\alpha$**. In **Appendix 5.3**, where we present a full sensitivity analysis for $\alpha$. This analysis shows: The **relative performance ranking** of the different methods (e.g., Struc-Emb variants vs. post-hoc variants) is **highly consistent across almost the entire range of $\alpha$ values.**
>
> Below, to further respond to the reviewer’s concern, we additionally provide a study on three graph datasets and **select the $\alpha$ over the validation set and report the test split result** on the selected $\alpha$. From the table below, we still observe **consistent benefit from Sturc-Emb variants** against post-hoc baselines even if we choose the semantic balancing coefficient not on the test set.
>
> | alpha value (Qwen 4B)                   | cora | citeseer | pubmed |
> |----------------------------------|------|----------|--------|
> | interpolation with mean neighbor | 0.06 | 0.66     | 0.88   |
> | interpolation with weighted mean | 0.06 | 0.66     | 0.88   |
> | interpolation with concat        | 0.48 | 0.66     | 0.82   |
> | interpolation with MP            | 0.10 | 0.48     | 0.88   |
> | interpolation with MP context    | 0.08 | 0.56     | 0.98   |
>
> | Acc score (Qwen 4B)       | cora  | citeseer | pubmed |
> |----------------------------------|-------|----------|--------|
> | interpolation with mean neighbor | 73.06 | 70.15    | 84.96  |
> | interpolation with weighted mean | 73.80 | 70.07    | 84.96  |
> | interpolation with concat        | 75.65 | 69.56    | 85.66  |
> | interpolation with MP            | 75.65 | 70.11    | 85.01  |
> | interpolation with MP context    | 74.72 | 70.46    | 84.99  |

---

### Official Review · Reviewer_xRbZ · 2025-11-01

**Soundness:** 3
**Presentation:** 3
**Contribution:** 3
**Rating:** 6
**Confidence:** 4

**Summary:**

This paper is motivated by the question of how to integrate structural information with the internal knowledge of LLM encoders to improve text embedding quality. It explores structure-aware text embedding by integrating structural information directly into the LLM’s internal encoding process. Through experiments, the paper shows the advantages of the two methods, Struc-Emb-Seq and Struc-Emb-Par. The paper is well structured and easy to follow. However, the lack of theoretical support and comprehensive experiments, such as fine-tuning and few-shot settings, limits its soundness.

**Strengths:**

1. This paper proposes structure-aware text embedding by integrating structural information directly into the LLM’s internal encoding process.
2. Through experiments, the paper shows the advantages of the two proposed methods, Struc-Emb-Seq and Struc-Emb-Par.
3. The paper is well structured and easy to follow.

**Weaknesses:**

1. The experimental results do not show a large advantage for these two methods; individual embeddings also show the best performance among the experiments.
2. This paper lacks an analysis of computational comparisons.
3. The paper focuses on zero-shot experimental results, lacking experiments such as fine-tuning and few-shot, which limits its soundness.
4. The captions of figures and tables lack clear notation and summarization of the results.

**Questions:**

1. Can this framework be evaluated with more extensive experiments, such as fine-tuning and few-shot learning, to strengthen its soundness?
2. What is the computational difference between LLMs and structure-aware models?

---

> ### Author Response · Authors · 2025-11-22
>
> We thank the reviewer for appreciating the idea of in-process structure-aware encoding, the experimental validation and structure of the paper. Below we respond to the questions raised by the reviewer.
>
> > 1). Not showing a large advantage of proposed method over individual embeddings (Weakness 1)
>
> We thank the reviewer for pointing out this confusion. Actually, our proposed Struc-Emb methods show **consistent and noticeable improvements over individual embeddings with semantic balancing** if comparing the results from the w.balancing part (lower half of the table) to the individual embedding performance. Without semantic balancing (w/balancing results), indeed we acknowledge that sometimes Struc-Embs alone can underperform individual embeddings when neighbor noise or distribution shifts dominate as we discussed in RQ1 point 3). This indeed underscores the need for semantic balancing to preserve the target information.
>
> > 2). Computational comparisons (Weakness 2 and Question 2)
>
> We include a computational comparison in Appendix 4 due to the limited space in the main text, where we observe that Struc-Emb-Par variants are close to individual encoding, with only slight overhead, and Struc-Emb-Par-Distill adds minimal cost over Struc-Emb-Par and Struc-Emb-Seq has a longer computation time in comparison.
>
> Below is the table recording the computation time of each method under the MuSiQue dataset when we vary the length of document and the number of relevant documents.
>
> | Musique degree | top5 100 | top10 100 | top5 500    | top5 1000   | top5 3000   | top5  fulldoc  |
> |------------------|----------|--------|--------|--------|--------|----------|
> | individual       |   0.0809   | 0.0809 | 0.2277 | 0.3665 | 0.8028 | 1.8679   |
> | concat           | 0.3636   | 1.717  | 1.8275 | 3.9052 | 14.2009| 32.4992  |
> | MP               | 0.0879   | 0.1343 | 0.3126 | 0.6217 | 2.281  | 4.4286   |
> | MP context       |  0.1103   | 0.1627 | 0.3396 | 0.6568 | 2.3257 | 4.5072   |
>
> > 3). Extension to fine-tuning and few-shot settings (Weakness 3 and Question 1)
>
> We thank the reviewer for pointing out the excellent further direction of this work. Just as what we discussed in the conclusion and future work, this paper is a first step preliminary study under zero-shot setting to verify the potential of in-process structure-aware encoding. The framework here is not limited to the zero-shot setting, and is directly applicable to further fine-tuning scenarios. And we agree that further fine-tuning or few-shot learning will definitely lead to improved performance since it can adapt the model to handle individually encoded parallel caches and better understand the balance between potentially noisy contextual segments to target segments.
>
> We focus on zero-shot setting as this paper is a first systematic study to investigate the native, "plug-and-play" capabilities and inherent challenges of in-process structural encoding. This zero-shot approach allowed us to isolate the effect of the structural context without the confounding variable of fine-tuning and reveal the inherent problems (like noise dominance, as discussed in Point 1), which led to our proposals for Context Distillation and Semantic Balancing. Also, as a preliminary step, the semantic balancing step with parameter $\alpha$ can effectively serve as an example of few-shot adaptation. As the table below shows, we still observe consistent benefits from Sturc-Emb variants against post-hoc baselines. While few examples might not be sufficient to finetune the embedding model. Ultimately, the next step is to further fine-tune the framework to unlock more potential regarding this type of structure-aware encoder.
>
> | Acc score (Qwen 4B)       | cora  | citeseer | pubmed |
> |----------------------------------|-------|----------|--------|
> | interpolation with mean neighbor | 73.06 | 70.15    | 84.96  |
> | interpolation with weighted mean | 73.80 | 70.07    | 84.96  |
> | interpolation with concat        | 75.65 | 69.56    | 85.66  |
> | interpolation with MP            | 75.65 | 70.11    | 85.01  |
> | interpolation with MP context    | 74.72 | 70.46    | 84.99  |
>
> 4). Captions of figures and tables (Weakness 4)
>
> We thank the reviewer for carefully reading through our tables and figures captions and raising this actionable feedback. We will further clarify the notations and add some result summary in the captions for better readability.

---

### Official Review · Reviewer_o8N3 · 2025-11-09

**Soundness:** 3
**Presentation:** 3
**Contribution:** 1
**Rating:** 2
**Confidence:** 4

**Summary:**

The authors present an approach to integrate structural information (such as links or citations) directly into the encoding process of pre-trained language models.
They propose two versions: Struc-Emb-Seq, which concatenates the target text with related texts   and Struc-Emb-Par, which uses parallel caches to incorporate the context more efficiently.
They also introduce two improvement techniques, Context Distillation and Semantic Balancing, aimed at reducing noise and preserving the meaning of the main text.
Experiments on several datasets show moderate improvements.

**Strengths:**

The idea that structured texts contain useful relationships is valid. The two proposed strategies, sequential and parallel, are well defined and easy to reproduce.
The analysis is systematic, as the authors test different types of datasets and provide a clear comparison with the baselines. The writing is clear, and the paper uses standard metrics and presents consistent results.

**Weaknesses:**

- Integrating structural context or linked documents into LLMs is not a new idea.
- The authors propose two well-implemented, but quite straightforward, approaches for adding context to the model. Concatenating texts or reusing parallel caches is not a significant conceptual innovation, but rather a simple variation in input processing.
- Maybe I missed it, but I could not find any analysis explaining why the use of structural context works or how it interacts with semantic representation; everything remains at the implementation level.
- The improvements are present but small, showing no clear qualitative leap—only marginal gains over the baselines.

The paper is well-executed and experimentally thorough, but lacks novelty. I would describe it as incremental work: it shows that certain practical choices (integrating structure during encoding) can yield slightly better results, but it does not advance the state of the art. It neither opens new research directions nor provides a theoretical framework. The paper is not poorly done, but it is not sufficiently innovative for this venue.

**Questions:**

It would be helpful if the authors could further stress the conceptual novelty of their approach in relation to previous studies.

---

> ### Author Response · Authors · 2025-11-22
>
> We thank Reviewer o8N3 for the feedback and the opportunity to clarify our paper's conceptual novelty and contributions. We understand the concern regarding novelty, and we respectfully clarify a critical distinction that defines our paper's contribution and separates it from prior work.
>
> > 1). Conceptual novelty of the paper in terms of the position and motivation (Weakness 1 and Question)
>
> We thank the reviewer for pointing out their confusion regarding the conceptual novelty and contribution of our paper, but we respectively disagree that our paper lacks novelty and being incremental to the field. Our work's innovation lies in a precise, fundamental distinction: we are the **first to systematically study the in-process integration of structural information directly into text embedding encoders**.
>
> Below we detail how this work is conceptually distinct from prior two categories of works:
>
> i). Comparison to previous works focusing on **generative tasks** that integrate contexts:
>
> First, we would like to highlight that prior works that involve integrating contexts or structures are mostly focusing on generative tasks. For instance, in RAG where the query can aggregate the information from all retrieved documents and answer the question [1, 2]. Or for the tasks on graph structured data, they project the graph features into the LLM’s token space to generate the answer [3, 4]. Indeed, the **task objective and model inductive bias can be fundamentally different** for embedding tasks and integrating structural contexts primarily for **generating general-purpose embedding is largely understudied**.
>
> ii). Comparison to previous **contextual embedding** works
>
> We agree that there are some embedding works that utilize and integrate the context before. But our work also significantly differ with them in the following two aspects:
>
> - **"In-Process" vs. "Post-Hoc" Integration:** Prior work on structure-aware embeddings almost exclusively relies on post-hoc aggregation given individually encoded embeddings from related segments. The aggregation can either be simple average pooling like [5, 6] or learnable [7, 8]. Or they encourage document pairs that are semantically relevant to be closer in the representation space [9]. In contrast, our "in-process" framework is a conceptual shift instead of just a practical design. It allows the model to leverage its internal, token-level attention mechanisms to **natively fuse structural context during the embedding's generation**, rather than operating on a lossy, pre-computed summary. This avoids the need to design or train complex, external aggregation functions that may not generalize well.
>
> [1. ]Wang, Yu, et al. "Knowledge graph prompting for multi-document question answering." Proceedings of the AAAI conference on artificial intelligence. Vol. 38. No. 17. 2024.
>
> [2]. Han, Haoyu, et al. "Retrieval-Augmented Generation with Graphs (GraphRAG)." CoRR (2025).
>
> [3]. Chen, Runjin, et al. "LLaGA: Large Language and Graph Assistant." International Conference on Machine Learning. PMLR, 2024.
>
> [4]. de Jong, Michiel, et al. "Fido: Fusion-in-decoder optimized for stronger performance and faster inference." Findings of the Association for Computational Linguistics: ACL 2023. 2023.
>
> [5]. Hansen, Casper, et al. "Contextual and sequential user embeddings for large-scale music recommendation." Proceedings of the 14th ACM Conference on Recommender Systems. 2020.
>
> [6]. Bendada, Walid, et al. "On the consistency of average embeddings for item recommendation." Proceedings of the 17th ACM Conference on Recommender Systems. 2023.
>
> [7]. Morris, John Xavier, and Alexander M. Rush. "Contextual Document Embeddings." The Thirteenth International Conference on Learning Representations.
>
> [8]. Zerveas, George, et al. "CODER: An efficient framework for improving retrieval through Contextual Document Embedding Reranking." Proceedings of the 2022 conference on empirical methods in natural language processing. 2022.
>
> [9]. Raman, Natraj, Sameena Shah, and Manuela Veloso. "Structure and semantics preserving document representations." Proceedings of the 45th International ACM SIGIR Conference on Research and Development in Information Retrieval. 2022.

---

> > ### Author Response · Authors · 2025-11-22
> >
> > > Continue from 1). Conceptual novelty of the paper in terms of the position and motivation (Weakness 1 and Question)
> >
> > - **Data-centric structural relations and broader scope:** The **contextual structure relation is explicitly obtained from real-world networks**, such as the wikipedia network and the online purchase network where the structure relations are predefined by the data and query/task independent in contrast to the induced KNN graphs [7, 11] or chunk relation within a document [10]. Also, previous works typically focus on one specific application domain like documents or for recommender systems. While we include a **variety of embedding tasks** in our paper ranging from documents for retrieval, products for recommendation or classification and question post for clustering to showcase the capability as producing general-purpose structure-ware embeddings.
> >
> > Therefore, we claim that our proposed "in-process" approach is a novel, general framework that opens up the potential of structure-aware encoding in a new angle that is **naturally compatible with latest LLM encoders**. Our paper provides the first systematic study of its viability, benchmarks its core methods, and analyzes the trade-offs for this new paradigm. We would be happy to discuss any further related work involving similar in-process, structure-aware encoding that we may have missed.
> >
> > [10]. Conti, Max, et al. "Context is Gold to find the Gold Passage: Evaluating and Training Contextual Document Embeddings." arXiv preprint arXiv:2505.24782 (2025).
> >
> > [11]. Yu, HongChien, Chenyan Xiong, and Jamie Callan. "Improving query representations for dense retrieval with pseudo relevance feedback." Proceedings of the 30th ACM International Conference on Information & Knowledge Management. 2021.
> >
> > > 2). Discussion of the method regarding its novelty and performance (Weakness 2)
> >
> > We thank the reviewer for carefully investigating our proposed two methods, but we would like to further restate our contribution in terms of the methods.
> >
> > - **Simple methods do not mean a lack of novelty:** First, as we stated above, previous contextual or structural embeddings, people all adopt post-hoc aggregation methods, we novelly integrate the contextual relation information into the LLM encoding process and **change fundamentally in the ways to get structure-aware embeddings**. The way we do in-process structure encoding is **simple but effective** as the first systematic study.
> > - **Struc-Emb-Par is non-trivial:** Also, Struc-Emb-Par (parallel caching) is not a simple reuse of caches. It is **tailored to effectively simultaneously aggregate information from multiple related segments guided by structure relations in the data**, which has not been explored in previous contextual embedding works. This requires non-trivial adaptations, including proper allocating positional encodings in a shared range to mitigate bias and modifying the attention mechanism to ground sparse attention according to structure relations in datasets. The "message-passing-like" aggregation of KV caches is a novel adaptation.
> > - **Systematic Study of Trade-offs:** Lastly, our primary goal was not just to propose a single SOTA algorithm. It was to establish this framework and provide the **first analysis of the fundamental trade-offs** between these two distinct in-process approaches. As we show, this choice is not trivial: **Struc-Emb-Seq** excels on noisy, moderate-length inputs but degrades on long texts, while **Struc-Emb-Par** scales robustly but is more susceptible to distractors. Also, we recognize the **importance of adding context distillation and semantic balancing** to mitigate the noisy context against target semantics.
> >
> > As stated, our contribution is not just a claim of uniform SOTA performance with rather complicated method, but the introduction, benchmarking, and systematic analysis of the in-process framework itself. We believe this framework, proposed simple but effective Struc-Emb methods and our initial analysis provide a strong foundation for future work.

---

> ### Author Response · Authors · 2025-11-22
>
> > 3). Analysis of why the use of structural context works or how it interacts with semantic representation (Weakness 3)
>
> We respectfully disagree that our paper "lacks analysis" and "remains at the implementation level." The core of our Results section is dedicated to moving beyond implementation to analyze how and why these methods work.
>
> - **Why it works (RQ1):** We analyze the conditions under which structural context is most beneficial. We show (in multi-hop QA and SportsFit) that gains are largest when text-only representations are "insufficient," quantitatively explaining why the context is needed.
> - **Interaction with Semantics (RQ2.1, RQ2.2):** We directly analyze the interaction between contextual information and target semantics.
>    - In **RQ2.2**, we perform a controlled study (MuSiQue) on the "distractor" effect of noisy context and how it impacts the embedding quality against the target semantics, which motivated our Context Distillation and Semantic Balancing proposals.
>    - In **RQ2.1 and Appendix 4**, we analyze how the Seq and Par pipelines interact differently with context order and length, showing Seq is order-sensitive while Par is more robust.
> - **Hyperparameter Analysis (Appendix 5.3):** We further analyze the reliance on structural context (via the semantic balancing hyperparameter) across different datasets, models, and encoding methods.
>
> > 4). Marginal gain over baselines (Weakness 4)
>
> We would like to respectively argue that our methods do not only present marginal gains over baselines like individual embeddings and post-hoc aggregations. Also, even sometimes we observe modest improvements in some cases, our analysis in RQ1 and RQ3 explain why it happens.
>
> - When comparing the **individual embeddings as baseline**:
>
> Our proposed Struc-Emb methods show **consistent and noticeable improvements over individual embeddings with semantic balancing** if comparing the results from the w.balancing part (lower half of the table) to the individual embedding performance. Without semantic balancing (w/balancing results), indeed we acknowledge that sometimes Struc-Embs alone can underperform individual embeddings when neighbor noise or distribution shifts dominate as we discussed in RQ1 point 3). This indeed underscores the need for semantic balancing to preserve the target information.
>
> Also, as discussed in RQ1, the gain from adding structural information is relatively small when the task is solvable with text alone (as in some graph classification tasks).
>
> - When comparing the **post-hoc aggregations as baseline**:
>
> Struc-Emb methods significantly outperform post-hoc methods if the **tasks require details** from the context. For instance, for multi-hop QA tasks in MuSiQue dataset, the post-hoc aggregation baselines show a significant gap with the top Struc-Emb method. Also, in the sportsfit dataset where there are on average 10 related items, Struc-Emb methods also consistently outperform the post-hoc baselines considerably since the summary of related embeddings may dilute signal and does not guarantee meaningful semantics especially with **increasing related segments**.
>
> On the other hand, as discussed in RQ3, when the number of related segments is very small (e.g. clustering, cora, citeseer, pubmed), the simple post-hoc aggregation is sufficient and won’t dilute the signal. Also, the gain over post-hoc aggregation is relatively marginal in STaRK-Amazon dataset with Hit@1 and MRR, since averaging embeddings can preserve the strongest signal most relevant to these metrics, but suffer on other metrics like Recall@n which depends on more comprehensive semantic aggregation.

---

### Official Review · Reviewer_xKdw · 2025-11-14

**Soundness:** 3
**Presentation:** 3
**Contribution:** 3
**Rating:** 8
**Confidence:** 2

**Summary:**

The paper introduces STRUC-EMB, a framework for generating structure-aware text embeddings by integrating relational information (such as hyperlinks, citations, or co-purchase links) directly into an LLM’s encoding process. It proposes two methods—sequential concatenation and parallel KV caching—and enhances them with context distillation and semantic balancing to handle noise and preserve target semantics. Across retrieval, clustering, classification, and recommendation tasks, these in-process methods consistently outperform text-only and post-hoc aggregation baselines, showing that directly encoding structural context yields more effective and scalable embeddings.

**Strengths:**

- Well-written paper and very pleasant to read.
- Comprehensive and carefully designed experiments.
- Investigating structure-aware encoding is both timely and genuinely valuable.

**Weaknesses:**

- The core findings are somewhat expected (e.g., structural information helps, and one must balance target embeddings against noisy context), though the proposed techniques—Context Distillation and Semantic Balancing—are thoughtful and appreciated.
- I might appreciate a deeper discussion or qualitative/mechanistic analysis of how post-hoc aggregation fails to capture low-level interactions.

Minor:
- Line 200, ’structurlly’

**Questions:**

- Apologies if I missed this, but could you clarify the experimental configurations used for the Mean Neighbour and Weighted Mean Neighbour baselines?

---

> ### Author Response · Authors · 2025-11-22
>
> We thank the reviewer xKdw for appreciating the value and timeliness of our topic, the experimental design and presentation of our paper. Below I will respond to some of the questions raised by the reviewer.
>
> > 1). Somewhat expected core findings (from weakness 1)
>
> We thank the reviewer for this observation. We agree that some of our findings, such as the benefit of structural information, align with intuition. However, it is surprisingly notable that no previous works have actually adopted **in-process structure-aware encoding**; prior methods have relied almost exclusively on **post-hoc aggregation** on top of related embeddings. We are the first study to systematically explore and evaluate this in-process approach.
>
> Also, our experiments revealed several **novel and non-trivial insights** that are not widely expected:
>
> - **Conditional Benefits:** We identify the specific conditions where structural context provides the most benefit. As shown in our results, the performance gap widens significantly when text-only embeddings are insufficient (e.g., when textural information is already sufficient or embeddings from weaker encoders), demonstrating when the added structural information is mostly effective.
> - **Pipeline Trade-offs:** We uncovered a critical trade-off between our two proposed pipelines. As discussed in our analysis, **Struc-Emb-Seq** (sequential) excels on noisy, moderate-length inputs but sees performance degrade on very long texts. Conversely, **Struc-Emb-Par** (parallel) scales robustly with long contexts and excels with high-signal context, but it is more susceptible to "distractor" noise. This trade-off provides a practical guide for implementation.
>
> > 2). Discussion in the failure of post-hoc aggregation (from weakness 2)
>
> Thank you for raising this clarification question in a deeper analysis of the post-hoc aggregation methods limitation. The failure of post-hoc aggregation, particularly its inability to capture "low-level interactions," stems from the following two aspects:
>
> - **Irreversible Information Loss:** Post-hoc methods (like mean-pooling) first compress each related segment into a **single, fixed-size embedding**. This "summarization" step is lossy and **irreversibly discards token-level granularity** and fine-grained facts. For instance, for multi-hop QA tasks in MuSiQue dataset that require retrieving specific and factual details from the context, the post-hoc aggregation baselines show a significant gap with the top Struc-Emb method.
> - **Signal Dilution:** The subsequent step in averaging these summary embeddings further **dilutes** the specific, low-level details needed for complex tasks especially when the number of related segments is large. Also, the summary of these embedding does not necessarily guarantee a meaningful semantic summary of corresponding related segments. This can be evidenced in the result from the Sportsfit dataset for item category classification where there are on average 10 related items, Struc-Emb methods also consistently outperform the post-hoc baselines considerably.
>
> In contrast, our **in-process methods** preserve this low-level information. By either sequentially concatenating (Struc-Emb-Seq) or co-processing the KV caches (Struc-Emb-Par), the target segment's encoding process can **directly attend to individual tokens** across the entire structural context. This allows the model to "look back" and aggregate the precise tokens needed to represent the target, rather than operating on diluted summaries.
>
> > 3). Experimental setting for mean neighbor and weighted mean neighbor (from question)
>
> Apologies for any ambiguity, below is a detailed description of how we perform mean neighbor and weighted mean neighbor as our baseline. First, we encode each structurally related segment (each neighbor) of the target segment individually and obtain these embeddings. Second, we are to compute the aggregated embedding of these structurally related segments. For the mean pooling, we just do a uniform average of these related embeddings and for the weighted mean neighbor, the average is weighted by the softmax-normalized cosine similarity with the target segment embedding. Finally, we interpolate the target segment embedding with the aggregated neighbor embedding from either mean-pooling or weighted mean-pooling. The interpolation follows the semantic balancing described in Sec. 3.3.

---

### Author Response · Authors · 2025-11-22

Dear Reviewers,

We sincerely appreciate your time and effort in providing us with valuable feedback to enhance our paper. We are grateful for your recognition of the in-process structure-ware encoding idea, simple and effective proposed method, comprehensive experimental evaluation and well organized paper. We will address each of these points in our detailed responses to individual reviewers.

---

> ### Author Response · Authors · 2025-12-03
>
> We are grateful to all our reviewers for their constructive feedback and insightful comments. Here we briefly summarize how our rebuttal addresses the main concerns and reinforces the strengths of the paper.
>
> First, we would like to highlight the consensus among reviewers on the strengths of our paper. The reviewers noted the significance of investigating structure-aware encoding within LLMs, the novelty of our proposed "in-process" paradigm and our paper is well-written and structured.
>
> Our rebuttals were targeted at some major concerns from some reviewers. In particular, we have carefully addressed the following aspects:
> >  1). Concern on the conceptual novelty of our paper (from reviewer o8N3) and on the comparisons to previous encode-time RAG/PRF works (from reviewer CJha):
> - **Response on the conceptual novelty of our paper**: To the best of our knowledge, we are the **first to systematically study the in-process integration of structural information directly into text embedding encoders**. This **differs from prior work on generative tasks**, where both objectives and inductive biases are fundamentally different, and where structural context is not primarily used to produce general-purpose embeddings. **Relative to prior contextual embedding methods**, our “In-Process” methods make a clear conceptual distinction with “post-hoc” baselines: we natively fuse structural context during embedding generation, and we explicitly leverage real-world, query-independent network relations, in contrast to RAG-style query-dependent retrieval of semantically similar documents.
>
> > 2). Concern on the key method innovation (from reviewer o8N3) and on the comparisons to previous encode-time RAG/PRF works (from reviewer CJha):
> - **Response on the method novelty:** Our methods are **simple but effective, aiming to fundamentally change the ways to get structure-aware embeddings**. In particular, Struc-Emb-Par is **tailored to effectively simultaneously aggregate information from multiple related segments guided by structure relations in the data, which includes non-trivial designs** such as positional encoding reallocation and attention modification. Also, our primary goal was not just to propose a single SOTA algorithm. It was to establish this framework and provide the **analysis of the fundamental trade-offs between these two distinct in-process approaches**.
>
> > 3). Concern on the improvements over baselines (from reviewer o8N3 and xRbZ)
> - **Response on the performance over baselines:** **When comparing the individual embeddings as baseline**, Our proposed Struc-Emb methods show consistent and noticeable improvements over individual embeddings with semantic balancing. **When comparing the post-hoc aggregations as baseline**, Struc-Emb methods significantly outperform post-hoc methods if the tasks require details from the context and with increasing related segments. In cases with **modest improvements, our analyses in RQ1 and RQ3 explain the behavior** (e.g., when structure adds less incremental information).
>
> > 4). Concern on the generalization beyond zero-shot setting (from reviewer xRbZ) and hyperparameter tuning beyond the oracle setting on the test split (from reviewer CJha)
> - **Response on the generalization beyond zero-shot setting:** The proposed framework is not limited to the zero-shot setting. This paper positions zero-shot as a first, controlled step to demonstrate the potential of in-process structure-aware encoding. We further performed hyperparameter tuning of the semantic balancing coefficient $\alpha$ on validation sets, as suggested by Reviewer xRbZ, which showcases few-shot capabilities and demonstrates robustness beyond the oracle setting on the test split (addressing Reviewer CJha).
>
> Finally, we have also clarified **computational costs** (from reviewer xRbZ), **baseline implementations** (from reviewer xKdw), and **detailed result interpretations** (from reviewer xKdw and o8N3) in the rebuttal.

---

### Meta-Review · Area_Chair_wZ44 · 2026-01-02

**Summary:**

This paper presents a framework for generating structure-aware text embeddings by integrating relational information (like hyperlinks, citations) directly into an LLM’s encoding process. It proposes two methods: simple sequential concatenation and an improved method with parallel caching. These two methods are improved by using a context distillation and semantic balancing to handle noise and preserve target semantics. Extensive zero-shot evaluations are proposed across retrieval, clustering, classification, and recommendation.

From the original reviews.

Reviewer xKdw find on the positive side the paper pleasant and well-written with comprehensive and carefully designed experiments. Investigating structure-aware encoding is both timely and genuinely valuable. On the other hand, the core findings are somewhat expected - ie that structural information helps- but the proposed techniques are appreciated (context distillation and semantic balancing), the paper lacks a discussion/qualitative analysis on how post-hoc aggregation fails to capture low-level interactions.

o8N3 finds the idea of using relationships of structured texts valid. The two strategies are well-defined and easy to reproduce. Analysis is systematic. Writing is clear with consistent results. On the other hand, the idea of integrating structural information is not new, the two strategies are somewhat straightforward looking more as a simple variation. Improvements are marginal. Novelty appears weak.

xRbZ identifies a strengths the integration of structural information directly into LLM's internal encoding, advantages of the two methods are shown in the experiments, paper well-structured and easy to follow. On the other hand: experimental gains are limited and individual embeddings show better performance in some experiments, lack of analysis of computational comparisons. The paper focuses on zero-shot experimental results, lacking experiments such as fine-tuning and few-shot, which limits its soundness. Notation and conclusion could be improved.

CJha identifies the following strengths: clear articulation with the two complementary design and principled discussion. Evaluations report consistent gains, methods are training-free inference. On the other hand, positioning wrt encode-time RAG/PRF is under-specified - it seems that there are strong connections with other papers, stricter selection of self union heuristics would strengthen claims.


Overall, the paper proposes interesting methods for generating structured-aware embeddings. Reviewers have appreciated the principle of the ideas even though the novelty, real in the sense that the method works internally wrt the LLM internal encoding process, is also discussed with respect to other works. Experimental evaluation is systematic and large but the method lacks comparison with other methods and paradigms and the results are not always that significant.
Paper is borderline, I have nevertheless the feeling that experimental evaluation could have been strengthened.
I propose rejection.

**Reviewer Concerns:**

For Reviewer xKdw, authors provided answers on all the weaknesses and question. In particular they argue that no method a priori relied on post-hoc aggregation at the internal encoding process of LLMs, plus the presence of novel insights.

For o8N3 they defend the novelty of the work, in particular the fact the fusion of structural context in during embedding generation (I'm not detailing the whole answer here) and the work is compatible with latest LLM encoders.

For xRbZ, they pointed the computation comparison in Appendix, and justify when the method is better than individual embeddings, insights on the potential of fine-tuning/few-shot is discussed.

For CJha have argued on the differences with other methods and precise their tuning.

**Reviewer Scores:**

Reviewer xKdw gave an 8 but with low confidence. I think he would have kept his score.

o8N3 gave a 2, it is difficult to evaluate if the reviewer would be fully convinced by the answers, maybe on the novelty of the method. Score could go not higher than 3/4 in my opinion.

xRbZ gave a 6, authors provided answers, it's not clear for me that the reviewer would have increased his score.

CJha gave a 6, again it not clear if the reviewer would have increased his score.

---

### Decision · Program_Chairs · 2026-01-26

Reject